# Single-shot General Hyper-parameter Optimization for Federated Learning

**Y. Zhou, P. Ram, T. Salonidis, N. Baracaldo, H. Samulowitz, H. Ludwig**
IBM Research
{yi.zhou, parikshit.ram}@ibm.com,
{tsaloni, baracald, samulowitz, hludwig}@us.ibm.com

## Abstract

We address the problem of hyper-parameter optimization (HPO) for federated learning (FL-HPO). We introduce **F**ederated **Lo**ss Su**R**face **A**ggregation (FLoRA), a general FL-HPO solution framework that can address use cases of tabular data and any Machine Learning (ML) model including gradient boosting training algorithms, SVMs, neural networks, among others and thereby further expands the scope of FL-HPO. FLoRA enables single-shot FL-HPO: identifying a single set of good hyper-parameters that are subsequently used in a *single* FL training. Thus, it enables FL-HPO solutions with minimal additional communication overhead compared to FL training without HPO. Utilizing standard smoothness assumptions, we theoretically characterize the optimality gap of FLoRA for any convex and non-convex loss functions, which explicitly accounts for the heterogeneous nature of the parties' local data distributions, a dominant characteristic of FL systems. Our empirical evaluation of FLoRA for multiple FL algorithms on seven OpenML datasets demonstrates significant model accuracy improvements over the baselines, and robustness to increasing number of parties involved in FL-HPO training.

## 1 Introduction

Traditional machine learning (ML) approaches require training data to be gathered at a central location where the learning algorithm runs. In real world scenarios, however, training data is often subject to privacy or regulatory constraints restricting the way data can be shared, used and transmitted. Examples of such regulations include the European General Data Protection Regulation (GDPR), California Consumer Privacy Act (CCPA), Cybersecurity Law of China (CLA) and HIPAA, among others. Federated learning (FL), first proposed in McMahan et al. (2017b), has recently become a popular approach to address privacy concerns by allowing collaborative training of ML models among multiple parties where each party can keep its data private.

**FL-HPO problem.** Despite the privacy protection FL brings along, there are many open problems in FL domain, one of which is hyper-parameter optimization for FL or FL-HPO (Kairouz et al., 2019; Khodak et al., 2021). Existing FL systems require a user (or all participating parties) to pre-set (agree on) multiple hyper-parameters (HPs) (i) for the model being trained (such as number of layers for neural networks or tree depth and number of trees in tree ensembles), (ii) for the FL algorithms, and (iii) for aggregation (if such hyper-parameters exist). Hyper-parameter optimization (HPO) for FL is important because the choice of HPs can have dramatic impact on model performance much like in traditional centralized ML (McMahan et al., 2017b).

While HPO has been widely studied in the centralized ML setting (Hutter et al., 2019), it comes with unique challenges in the FL setting. First, existing HPO techniques often make use of the entire dataset, which is not available centrally in FL. Secondly, they need to train many models for a large number of HP configurations which is prohibitively expensive in terms of communication and training time in FL settings; training a single model already has a high communication overhead (Kairouz et al., 2019). Thirdly, one important challenge that has not been adequately explored in FL-HPO literature is support for tabular data, which are widely used in enterprise settings, such as financial services and other traditional industries, preferring traditional models with some explainability (Ludwig et al., 2020). Although a few approaches have been recently proposed for FL-HPO, they focus on handling

HPO using personalization techniques (Khodak et al., 2021) and neural networks (Khodak et al., 2020). To the best of our knowledge, there is no FL-HPO approach to train non-neural network models, such as gradient boosted decision trees (Friedman, 2001) (e.g., XGBoost (Chen & Guestrin, 2016)) that are particularly common in the enterprise setting, even though there are existing FL algorithms for such models (Li et al., 2020; Ong et al., 2020). This leads to our motivating question:

> Can we develop a FL-HPO scheme that performs HPO for any ML model in a FL environment without significantly increasing the already-high communication overhead of FL?

In this paper, we address the aforementioned challenges of FL-HPO and our motivating question. We focus on the problem where the model HPs are shared by all parties and we seek a set of HPs and train a single model that is used by all parties. Our motivating question leads to four further requirements that make the problem challenging: (**C1**) To **perform** FL-HPO **with any ML model**, we *cannot make any assumption* that two models with different HPs can perform some "weight-sharing", allowing our solution to be applied beyond fixed architecture neural networks. (**C2**) To be **general across ML models**, we *do not assume the ability to perform "multi-fidelity" HPO* to reduce the communication overhead of FL-HPO (see discussion in Appendix A.1). (**C3**) To **avoid increasing the FL communication overhead**, we seek to *perform "single-shot" FL-HPO*, where we can perform FL-HPO while requiring *only a single* FL model training. (**C4**) To **be applicable to FL with data heterogeneity**, we *cannot assume that parties have independent and identically distributed (IID) data*.

**Contributions.** Given the above FL-HPO problem setting, we make the following contributions:

- (§2) We present a novel framework **F**ederated **Lo**ss Su**R**face **A**ggregation (FLoRA) that leverages meta-learning techniques enabling **asynchronous local** HPOs on each party to perform single-shot HPO for the global FL-HPO problem.
- (§2.3) We provide theoretical guarantees for the set of HPs selected by FLoRA covering both IID and Non-IID cases regardless of the convexity of the loss function. To the best of our knowledge, this is the first rigorous theoretical analysis for FL-HPO problem and also the first optimality gap constructed in terms of the estimated loss given a target distribution.
- (§3) We evaluate FLoRA on the FL-HPO of Histogram based Gradient Boosted Decision Trees (HGB), Support Vector Machines (SVM) and Multi-layered Perceptrons (MLP) on seven classification datasets from OpenML (Vanschoren et al., 2013), highlighting (i) its performance relative to baselines, and (ii) the effect of data heterogeneity.

In Figure 1, we present a snapshot of our empirical results which highlights the communication overhead reduction we achieve from FLoRA while producing higher quality models. As baselines, we directly adopt an existing centralized HPO scheme that requires federated training of multiple models and term this a "multi-shot" FL-HPO baseline. The figure also shows a "single-shot" baseline that uses curated HPs (described in §3), and FLoRA is also single-shot. Figure 1 shows that the "multi-shot" approach requires a significantly large number of FL model trainings (39 for MLP and 24 for HGB) and hence more communication to find a HP that matches the performance of the HP found by FLoRA, highlighting the efficiency and effectiveness of FLoRA. This result demonstrates that FLoRA is a FL-HPO scheme that works with any ML model (HGB, MLP, etc), providing competitive performance without significantly increasing the communication overhead of FL by only requiring a *single* FL model training.

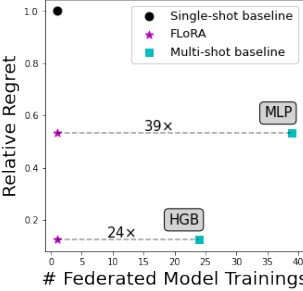

Figure 1: Communication overhead savings of FLoRA compared to "multi-shot" FL-HPO for the same level of performance. We are considering 2 FL-HPO problems on the Electricity dataset (with HGB and MLP). We use the *relative regret* (defined in §3) of each scheme as the performance metric (*lower is better*), where a regret of 1 denotes performance of the single-shot baseline while a regret of 0 implies optimal performance. FLoRA and the single-shot baseline require a single federated model training.

## 1.1 RELATED WORK

**Performance optimization of FL systems.** One of the main challenges in FL is achieving high accuracy with low communication overhead. FedAvg (McMahan et al., 2017a) is a predominant FL algorithm and several optimization schemes build on it. Initially, communication optimizations

included performing multiple stochastic gradient descent (SGD) local iterations at the clients and randomly selecting a small subset of the clients to compute and send updates to the server. Subsequently, compression techniques were used to minimize the size of model updates to the server. The accuracy and communication overhead of these techniques are sensitive to their HPs (McMahan et al., 2017a).

FL-HPO **of neural networks.** Recent optimization approaches adapt HPs such as the local learning rate at each client (Koskela & Honkela, 2019; Mostafa, 2019; Reddi et al., 2020), the number of local SGD epochs (Wang et al., 2019). Dai et al. (2020; 2021) address Federated Bayesian Optimization. Although using HPO with multiple HPs, the problem setup is quite different than FL: they focus on a single party using information from other parties to accelerate its own Bayesian Optimization, instead of building a model for all parties. Federated Neural Architecture Search (FNAS) approaches search for architectural HPs of CNN models by running locally NAS algorithms and then aggregating the NAS architecture weights and model weights using FedAvg (He et al., 2020; Garg et al., 2020; Xu et al., 2020). These approaches have shown empirical gains but lack theoretical analysis. Inspired from the NAS technique of weight-sharing, Khodak et al. (2020; 2021) proposed FedEx, a FL-HPO framework to accelerate a centralized HPO procedure, i.e., successive halving algorithm (SHA), for many SGD-based FL algorithms. FedEx focuses on building personalized models for parties by tuning local HPs of the parties. They provide theoretical guarantee only for a special case of tuning a single HP, the local learning rate, in an online convex optimization setting. To optimize global HPs, FedEx requires multiple rounds of communication, and hence is not single-shot FL-HPO (Khodak et al., 2020; 2021). Note that these above techniques are multi-shot, leveraging both the idea of weight-sharing and multi-fidelity HPO for improving the communication efficiency of FL-HPO. The FedHPO-B benchmark (Wang et al., 2022) primarily focuses on neural-networks and multi-fidelity FL-HPO and is useful to evaluate the above schemes across various problems.

**Need for** FL-HPO **of tabular data models.** As most existing FL-HPO approaches focus on SGD-based algorithms and neural networks, one major limitation they share is that they do not support tree-based models, such as gradient boosted trees (Friedman, 2001), a popular model for enterprise setting. These models provide explanability for predictions which is required for financial and healthcare FL use-cases. As laid out in a policy paper by the OECD, numerous regulations of member countries govern the use of analytics and data (OECD, 2021): GDPR, for example, requires decision-making models for financial services and insurance to be explainable, which is mostly achieved using traditional models such as decision tree variants (Goodman & Flaxman, 2017). Outside consumer finance, governance rules require explanability of portfolio and risk management for auditing purposes (Gensler & Bailey, 2020). Again, DNNs (deep neural networks) are not satisfactory from a current regulatory point of view and, thus, the financial services and insurance sectors rely on more explainable models (such as tree-based ones), also in federation.

**This paper.** Our framework improves on the above approaches in several ways, summarized also in Table 1. **(1) It is more general**, as it can tune multiple HPs and is applicable to non SGD-training settings such as gradient boosting trees. This is achieved by treating FL-HPO as a *black-box HPO problem* (as opposed to *grey-box HPO* where we can leverage techniques such as weight-sharing and multi-fidelity), which has been addressed in centralized HPO literature using grid search, random search (Bergstra & Bengio, 2012) and Bayesian Optimization approaches (Shahriari et al., 2016). The key challenge is the requirement to perform computationally intensive evaluations on a large number of HPO configurations, where each evaluation involves training a model and scoring it on a validation dataset. In the distributed FL setting this problem is exacerbated because validation sets are local to the parties and each FL training/scoring evaluation is communication intensive. Therefore a brute force application of centralized black-box HPO approaches that select HPs in an outer loop and proceed with FL training evaluations is not feasible. **(2) It yields minimal HPO communication overhead.** This is achieved by building a loss surface from local asynchronous HPO at the parties that yields a single optimized HP configuration used to train a global model with a single FL training. **(3) It is the first that theoretically characterizes the optimality gap in an** FL-HPO **setting**, for the case we focus in this paper: creating a global model by tuning multiple global HPs without accessing global validation dataset (as opposed to existing work either optimizing wrt parties' validation datasets or assuming access to a global validation dataset during HPO).

Table 1: Positioning of our proposed framework FLoRA against existing literature. SS: single-shot. MF: multi-fidelity. WS: weight-sharing. $\boldsymbol{\theta}_G$: global model HPs. $\boldsymbol{\phi}$: aggregator HPs. See §2. †: FedHPO-B is a benchmarking suite and not an algorithm, and the properties correspond to the problems in the suite.

| Method | Any ML model | SS | No MF req. | No WS req. | Handles $\boldsymbol{\theta}_G$ | Handles $\boldsymbol{\phi}$ | Black-box |
|---|---|---|---|---|---|---|---|
| Black-box HPO | ✓ | ✗ | ✓ | ✓ | ✓ | NA | ✓ |
| Grey-box HPO | ✗ | ✗ | ✗ | ✗ | ✓ | NA | ✗ |
| FNAS | ✗ | ✗ | ✗ | ✗ | ✓ | NA | ✗ |
| FedEx | ✗ | ✗ | ✗ | ✗ | ✓ | ✓ | ✗ |
| FLoRA (Ours) | ✓ | ✓ | ✓ | ✓ | ✓ | ✗ | ✓ |
| FedHPO-B$^\dagger$ | ✗ | ✗ | ✗ | - | ✓ | ✗ | ✗ |

## 2 METHODOLOGY

In the centralized ML setting, we would consider a model class $\mathcal{M}$ and its corresponding learning algorithm $\mathcal{A}$ parameterized collectively with HPs $\boldsymbol{\theta} \in \boldsymbol{\Theta}$, and given a training set $D$, we can learn a single model $\mathcal{A}(\mathcal{M}, \boldsymbol{\theta}, D) \to m \in \mathcal{M}$. Given some predictive loss $\mathcal{L}(m, D')$ of any model $m$ scored on some holdout set $D'$, the centralized HPO problem can be stated as

$$\min_{\boldsymbol{\theta} \in \boldsymbol{\Theta}} \mathcal{L}(\mathcal{A}(\mathcal{M}, \boldsymbol{\theta}, D), D'). \tag{2.1}$$

In the most general FL setting, we have $p$ parties $P_1, \ldots, P_p$ each with their private local training dataset $D_i, i \in [p] = \{1, 2, \ldots, p\}$. Let $D = \cup_{i=1}^{p} D_i$ denote the aggregated training dataset and $\overline{D} = \{D_i\}_{i \in [p]}$ denote the set of per-party datasets. Each model class (and corresponding learning algorithm) is parameterized by global HPs $\boldsymbol{\theta}_G \in \boldsymbol{\Theta}_G$ shared by all parties and per-party local HPs $\boldsymbol{\theta}_L^{(i)} \in \boldsymbol{\Theta}_L, i \in [p]$ with $\boldsymbol{\Theta} = \boldsymbol{\Theta}_G \times \boldsymbol{\Theta}_L$. FL systems usually include an aggregator which introduces an additional set of HPs $\boldsymbol{\phi} \in \boldsymbol{\Phi}$. Finally, we have a FL algorithm $\mathcal{F}$

$$\mathcal{F}\left(\mathcal{M}, \boldsymbol{\phi}, \boldsymbol{\theta}_G, \{\boldsymbol{\theta}_L^{(i)}\}_{i \in [p]}, \mathcal{A}, \overline{D}\right) \to m \in \mathcal{M}, \tag{2.2}$$

which takes as input all the relevant HPs and per-party datasets and generates a model. In this case, the FL-HPO problem can be stated in the two following ways depending on the desired goals: (i) Ideally, for a global holdout/validation dataset $D'$ (possibly from the same distribution as the aggregated dataset $D$), the target problem is:

$$\min_{\boldsymbol{\phi} \in \boldsymbol{\Phi}, \boldsymbol{\theta}_G \in \boldsymbol{\Theta}_G, \boldsymbol{\theta}_L^{(i)} \in \boldsymbol{\Theta}_L, i \in [p]} \mathcal{L}\left(\mathcal{F}\left(\mathcal{M}, \boldsymbol{\phi}, \boldsymbol{\theta}_G, \{\boldsymbol{\theta}_L^{(i)}\}_{i \in [p]}, \mathcal{A}, \overline{D}\right), D'\right). \tag{2.3}$$

(ii) An alternative target problem would involve per-party holdout datasets $D_i', i \in [p]$ as follows:

$$\min_{\boldsymbol{\phi} \in \boldsymbol{\Phi}, \boldsymbol{\theta}_G \in \boldsymbol{\Theta}_G, \boldsymbol{\theta}_L^{(i)} \in \boldsymbol{\Theta}_L, i \in [p]} \mathsf{Agg}\left(\left\{\mathcal{L}\left(\mathcal{F}\left(\mathcal{M}, \boldsymbol{\phi}, \boldsymbol{\theta}_G, \{\boldsymbol{\theta}_L^{(i)}\}_{i \in [p]}, \mathcal{A}, \overline{D}\right), D_i'\right), i \in [p]\right\}\right), \tag{2.4}$$

where $\mathsf{Agg} : \mathbb{R}^p \to \mathbb{R}$ is some aggregation function (such as average or maximum) that scalarizes the $p$ per-party predictive losses.

Contrasting problem (2.1) to problems (2.3) & (2.4), we can see that the FL-HPO is significantly more complicated than the centralized HPO problem. In the ensuing presentation, we focus on problem (2.3) although our proposed single-shot FL-HPO scheme can be applied and evaluated for problem (2.4). We simplify the FL-HPO problem in the following ways: (i) we assume that there is no personalization so there are no per-party local HPs $\boldsymbol{\theta}_L^{(i)}, i \in [p]$, (ii) we only focus on the model class HPs $\boldsymbol{\theta}_G$, deferring HPO for aggregator HPs $\boldsymbol{\phi}$ for future work as many of them are set based on the communication and computational resources available in the FL system and cannot be directly optimized with regards to some predictive performance metrics, and (iii) we assume there is a global holdout/validation set $D'$ which is only used to evaluate the final global model's performance but *can not be accessed* during HPO process. And parties can only access their own private training $D_i$ and validation $D_i'$ sets. Hence the problem we will study is stated as for a fixed aggregator HP $\boldsymbol{\phi}$:

$$\min_{\boldsymbol{\theta}_G \in \boldsymbol{\Theta}_G} \mathcal{L}\left(\mathcal{F}\left(\mathcal{M}, \boldsymbol{\phi}, \boldsymbol{\theta}_G, \mathcal{A}, \overline{D}\right), D'\right). \tag{2.5}$$

This problem appears similar to the centralized HPO problem (2.1). However, note that the main challenges in (2.5) is (i) the need for a federated training for each set of HPs $\boldsymbol{\theta}_G$, and (ii) the need to evaluate the trained model on the global validation set $D'$ (which is usually not available in usual FL-HPO setting). Hence it is not practical (from a communication overhead and functional perspective) to apply existing off-the-shelf HPO schemes to problem (2.5). In the subsequent discussion, for simplicity purposes, we will use $\boldsymbol{\theta}$ to denote the global HPs, dropping the "$G$" subscript.

## 2.1 Leveraging local HPOs

While it is possible but extremely expensive to apply off-the-shelf HPO solvers (such as Bayesian Optimization (BO) (Shahriari et al., 2016), Hyperopt (Bergstra et al., 2011), etc.), we wish to understand how we can leverage local and asynchronous HPOs in each of the parties. We begin with a simple but intuitive hypothesis underlying various meta-learning schemes for HPO (Vanschoren, 2018; Wistuba et al., 2018; Ram, 2022): *if a HP configuration θ has good performance for all parties independently, then θ is a strong candidate for federated training.*

With this hypothesis, we present our proposed FLoRA in Algorithm 1. In this scheme, we allow each party to perform HPO locally and asynchronously with some adaptive HPO scheme such as BO (line 3).

---

**Algorithm 1** FL-HPO with FLoRA

1: **Input:** $\Theta, \mathcal{M}, \mathcal{A}, \mathcal{F}, \{(D_i, D_i')\}_{i \in [p]}, T$
2: **for** each party $P_i, i \in [p]$ **do**
3:    Run HPO to generate $T$ (HP, loss) pairs

$$E^{(i)} = \left\{ (\boldsymbol{\theta}_t^{(i)}, \mathcal{L}_t^{(i)}), t \in [T] \right\}, \qquad (2.6)$$

$$\boldsymbol{\theta}_t^{(i)} \in \Theta, \mathcal{L}_t^{(i)} := \mathcal{L}(\mathcal{A}(\mathcal{M}, \boldsymbol{\theta}_t^{(i)}, D_i), D_i').$$

4: **end for**
5: Collect all $E = \{E^{(i)}, i \in [p]\}$ in aggregator
6: Generate a unified loss surface $\widehat{\ell} : \Theta \to \mathbb{R}$ using $E$
7: Select best HP candidate

$$\widehat{\boldsymbol{\theta}}^\star \leftarrow \arg\min_{\boldsymbol{\theta} \in \Theta} \widehat{\ell}(\boldsymbol{\theta}). \qquad (2.7)$$

8: Invoke federated training $m \leftarrow \mathcal{F}(\mathcal{M}, \widehat{\boldsymbol{\theta}}^\star, \mathcal{A}, \overline{D})$
9: **Output:** FL model $m$.

---

Then, at each party $i \in [p]$, we collect all the attempted $T$ HPs $\boldsymbol{\theta}_t^{(i)}, t \in [T] = \{1, 2, \ldots, T\}$ and their corresponding predictive loss $\mathcal{L}_t^{(i)}$ into a set $E^{(i)}$ (line 3, equation (2.6)). Then these per-party sets of (HP, loss) pairs $E^{(i)}$ are collected at the aggregator (line 5). This operation has at most $O(pT)$ communication overhead (note that the number of HPs are usually much smaller than the number of columns or number of rows in the per-party datasets). These sets are then used to generate an aggregated loss surface $\widehat{\ell} : \Theta \to \mathbb{R}$ (line 6) which will then be used to make the final single-shot HP recommendation $\widehat{\boldsymbol{\theta}}^\star \in \Theta$ (line 7) for the federated training to create the final model $m \in \mathcal{M}$ (line 8). We will discuss the generation of the aggregated loss surface in detail in §2.2. Before that, we briefly want to discuss the motivation behind some of our choices in Algorithm 1.

**Remarks.** Using adaptive HPO schemes instead of non-adaptive schemes (such as random search or grid search) allows us to efficiently approximate the local loss surface more accurately (and with more certainty) in regions of the HP space where the local performance is favorable instead of trying to approximate the loss surface well over the complete HP space. This has advantages both in terms of computational efficiency and loss surface approximation. Moreover, each party executes HPO asynchronously, without coordination with HPO results from other parties or with the aggregator. This is in line with our objective to minimize communication overhead. Although there could be strategies that involve coordination between parties, they could involve many rounds of communication. Our experimental results show that this approach is effective for the datasets we evaluated for.

## 2.2 Loss surface aggregation

Given the sets of (HP, loss) pairs $E^{(i)} = (\boldsymbol{\theta}_t^{(i)}, \mathcal{L}_t^{(i)}), i \in [p], t \in [T]$ at the aggregator, we wish to construct a loss surface $\widehat{\ell} : \Theta \to \mathbb{R}$ that best emulates the (relative) performance loss $\widehat{\ell}(\boldsymbol{\theta})$ we would observe when training the model on $\overline{D}$. Based on our hypothesis, we want the loss surface to be such that it would have a relatively low $\widehat{\ell}(\boldsymbol{\theta})$ if $\boldsymbol{\theta}$ has a low loss for all parties simultaneously. However, because of the asynchronous and adaptive nature of the local HPOs, for any HP $\boldsymbol{\theta} \in \Theta$, we would not have the corresponding losses from all the parties. For that reason, we will model the loss surfaces using regressors that try to map any HP to their corresponding loss. We present four ways of constructing such loss surfaces, and we also briefly summarize them in Table 2.

The most straightforward way to construct such a loss surface is to merge all the per-party sets $E^{(i)}$ to get a single set $E = \cup_{i \in [p]} E^{(i)}$ and use it to train a regressor $f : \Theta \to \mathbb{R}$ (such as a Random Forest Regressor (Breiman, 2001)) using the HPs $\boldsymbol{\theta}$ as the covariates and the corresponding loss as the dependent variable. Then we can define the loss surface as this *single global model* or **SGM** $\widehat{\ell}(\boldsymbol{\theta}) := f(\boldsymbol{\theta})$. However, this loss surface is extremely optimistic, assigning a low loss to a HP if it had a low loss estimate on any one of the parties, making it unsuitable in the presence of data heterogeneity. We can leverage uncertainty quantification $u : \Theta \to \mathbb{R}_+$ around regressor predictions to get a loss surface $\widehat{\ell}(\boldsymbol{\theta}) := f(\boldsymbol{\theta}) + \alpha u(\boldsymbol{\theta})$ for some $\alpha > 0$ – the *single global model with uncertainty*

or **SGM+U**. This would improve the robustness of SGM by penalizing parts of the HP space which were not well explored by all parties' local HPOs.

Instead of merging the per-party $E^{(i)}$, we can also use them to train a per-party *local regressor model* $f^{(i)} : \Theta \to \mathbb{R}$ and use their ensemble as the loss surface. One way is to use the *average of the per-party local models* or **APLM** as the loss surface $\hat{\ell}(\boldsymbol{\theta}) := {}^1/p \sum_{i \in [p]} f^{(i)}(\boldsymbol{\theta})$. This is less optimistic than SGM and provides some level of robustness in the presence of non-IID heterogeneous per-party distributions since it will assign a low loss for a HP only if its average across all per-party regressors is low, which implies that most parties observed a relatively low loss around this HP. An even more robust loss surface would be the *maximum of the per-party local models* or **MPLM**

Table 2: **Loss surfaces:** $f : \Theta \to \mathbb{R}$ is the global loss surface generated using the aggregated set $\cup_{i \in [p]} E^{(i)}$ of the per-party set of loss pairs $E^{(i)}$ from each party $i \in [p]$. $u : \Theta \to \mathbb{R}_+$ is an uncertainty model generated using the aggregated set $\cup_{i \in [p]} E^{(i)}$ and $\alpha > 0$ is a constant. $f_i : \Theta \to \mathbb{R}$ for any $i \in [p]$ is the per-party loss surface generated using the party's loss pairs $E^{(i)}$.

| Surface | $\hat{\ell}(\theta) :=$ | Optimism | Non-IID |
|---------|---------|----------|---------|
| SGM | $f(\theta)$ | High | ✗ |
| SGM+U | $f(\theta) + \alpha \cdot u(\theta)$ | Medium | Partial |
| MPLM | $\max_{i \in [p]} f_i(\theta)$ | Low | ✓ |
| APLM | ${}^1/p \sum_{i \in [p]} f_i(\theta)$ | Medium | ✓ |

$\hat{\ell}(\boldsymbol{\theta}) := \max_{i \in [p]} f^{(i)}(\boldsymbol{\theta})$ which would only assign a low loss to a HP only if it has low loss estimate across all parties, making it extremely capable of handling data heterogeneity (as we will also highlight in our empirical evaluations). We discuss these loss surfaces in detail in Appendix B. In §2.3, we theoretically quantify the performance guarantees for MPLM and APLM, and in §3, we empirically evaluate all these loss surfaces.

## 2.3 OPTIMALITY ANALYSIS

We now rigorously analyze the sub-optimality of the HP selected by FLoRA. We are interested in providing a bound for the following *optimality gap*:

$$\mathcal{G} := \tilde{\ell}(\widehat{\boldsymbol{\theta}}^\star, \mathcal{D}) - \tilde{\ell}(\boldsymbol{\theta}^\star, \mathcal{D}), \quad \text{where} \quad \boldsymbol{\theta}^\star \in \arg\min_{\boldsymbol{\theta} \in \boldsymbol{\Theta}} \tilde{\ell}(\boldsymbol{\theta}, \mathcal{D}). \quad (2.8)$$

Here $\tilde{\ell}(\boldsymbol{\theta}, \mathcal{D})$ is an estimate of the true loss $\ell(\boldsymbol{\theta}, \mathcal{D}) := \mathbb{E}_{(x,y) \sim \mathcal{D}} \mathcal{L}(\mathcal{A}(\boldsymbol{\theta}, \overline{D}), (x, y))$ (see Definition C.1) given some validation set $D'$ sampled from $\mathcal{D}$, which is the model performance metric during evaluation and/or inference time. Recall that $\widehat{\boldsymbol{\theta}}^\star$ selected by FLoRA is defined as in (2.7), and $\theta^*$ denotes the optimal HP given by $\tilde{\ell}$ for a desired data distribution $\mathcal{D}$ we want to learn.

We present our main results in Theorem 2.1. Informally, we show how to bound the optimality gap by picking the 'worst-case' HP setting that maximizes the combination of Wasserstein distances of the local data distributions and actual quality of local HPO approximation across parties. The more precise theorem statement and its proof with formal discussion of technical definitions and assumptions can be found in Appendix C.

**Theorem 2.1.** *Suppose that the loss estimate $\tilde{\ell}$ and the unified loss surface $\hat{\ell}$ are Lipschitz continuous. Consider the optimality gap $\mathcal{G}$ defined in (2.8), where $\widehat{\boldsymbol{\theta}}^\star$ is selected by FLoRA with each party $i \in [p]$ collecting $T$ (HP, loss) pairs $\{(\boldsymbol{\theta}_t^{(i)}, \mathcal{L}_t^{(i)})\}_{t \in [T]}$ during the local HPO run. For a desired data distribution $\mathcal{D} = \sum_{i=1}^p w_i \mathcal{D}_i$, where $\{\mathcal{D}_i\}_{i \in [p]}$ are the sets of parties' local data distributions and $w_i \in [0, 1], \forall i \in [p]$, we have*

$$\mathcal{G} \leq \max_{\boldsymbol{\theta} \in \bar{\boldsymbol{\Theta}}} \sum_{i \in [p]} C_{\boldsymbol{\alpha}} \left\{ C_\beta \sum_{j \in [p], j \neq i} w_j \mathcal{W}_1(\mathcal{D}_j, \mathcal{D}_i) + C_{\tilde{L}, \hat{L}_i} \min_{t \in [T]} d(\boldsymbol{\theta}, \boldsymbol{\theta}_t^{(i)}) + \delta_i \right\}. \quad (2.9)$$

*In particular, when all parties have i.i.d. local data distributions, (2.9) reduces to*

$$\mathcal{G} \leq \max_{\boldsymbol{\theta} \in \bar{\boldsymbol{\Theta}}} \sum_{i \in [p]} C_{\boldsymbol{\alpha}} \left\{ C_{\tilde{L}, \hat{L}_i} \min_{t \in [T]} d(\boldsymbol{\theta}, \boldsymbol{\theta}_t^{(i)}) + \delta_i \right\}.$$

*Here $C_{\boldsymbol{\alpha}}$, $C_\beta$ and $C_{\tilde{L}, \hat{L}_i}$ are constants only related to the unified loss surface and Lipschitz-ness, respectively, $\mathcal{W}_1(\cdot, \cdot)$ and $d(\cdot, \cdot)$ are distance metrics defined over data distribution and hyper-parameter space $\boldsymbol{\Theta}$, respectively, and $\delta_i$ is the maximum per sample training error for the local loss surface $\hat{\ell}_i$, i.e., $\delta_i = \max_t |\mathcal{L}_t^{(i)} - \hat{\ell}_i(\boldsymbol{\theta}_t^{(i)})|$.*

There are several interesting observations from Theorem 2.1: (i) The first term in our bound (2.9) characterizes the errors incurred by parties' data heterogeneity, measured via the 1-Wasserstein

Table 3: Comparison of different loss surfaces (the 4 rightmost columns) for FLoRA relative to the baseline for single-shot 3-party FL-HPO in terms of the *relative regret* (lower is better).

| Aggregate | ML Method | SGM | SGM+U | MPLM | APLM |
|---|---|---|---|---|---|
| Regret | HGB | [0.30, 0.47, 0.68] | [0.27, 0.54, 0.64] | [0.25, 0.43, 0.67] | [0.25, 0.50, 0.65] |
| Inter-quartile range | SVM | [0.04, 0.38, 1.11] | [0.04, 0.48, 1.07] | [0.38, 0.91, 2.41] | [0.23, 0.54, 0.76] |
| | MLP | [0.36, 0.80, 0.97] | [0.48, 0.99, 1.01] | [0.47, 0.89, 1.00] | [0.46, 0.79, 0.95] |
| | Overall | [**0.22**, **0.53**, 0.97] | [0.32, 0.55, 1.01] | [0.36, 0.61, 0.99] | [0.36, 0.57, **0.79**] |
| FLoRA | HGB | 6/0/1 | 6/0/1 | 7/0/0 | 7/0/0 |
| Wins/Ties/Losses | SVM | 4/0/2 | 4/0/2 | 3/0/3 | 5/0/1 |
| | MLP | 6/0/1 | 4/1/2 | 5/1/1 | 6/0/1 |
| | Overall | 16/0/4 | 14/1/5 | 15/1/4 | **18/0/2** |
| Wilcoxon Signed-Rank Test | HGB | (26, 0.02126) | (27, 0.01400) | (28, 0.00898) | (28, 0.00898) |
| 1-sided | SVM | (18, 0.05793) | (17, 0.08648) | (9, 0.62342) | (15, 0.17272) |
| (statistic, p-value) | MLP | (21, 0.11836) | (15, 0.17272) | (18, 0.05793) | (24, 0.04548) |
| | Overall | (174, 0.00499) | (164, 0.00272) | (141, 0.03206) | (**183.5, 0.00169**) |

distance (Villani, 2021) – the magnitude of Non-IIDness in a FL system; we can see it vanish under the IID setting. (ii) The last two terms measure the quality of the local HPO approximation, which can be reduced if a good loss surface is selected. For example, if we use non-parametric regression models as the loss surfaces, the per-sample training error can be arbitrarily small (that is $\delta_i \approx 0$), but at the cost of increasing $\widehat{L}_i$ for $\widehat{\ell}_i$. (iii) The $\min_{t \in [T]} d(\boldsymbol{\theta}, \boldsymbol{\theta}_t^{(i)})$ term indicates that the optimality gap depends only on the HP trials $\boldsymbol{\theta}_t^{(i)}$ that are closest to the optimal HP setting. (iv) If we assume each party's training dataset $D_i$ is of size $n_i$ sampled as $D_i \sim \mathcal{D}_i^{n_i}$, we can view $w_i = \frac{n_i}{n}$ where $n = \sum_{i=1}^p n_i$, i.e., with probability $w_i$ the desired data distribution $\mathcal{D}$ is sampled from $\mathcal{D}_i$.

Now we would like to compare our theoretical results with existing analyses such as Khodak et al. (2021) and He et al. (2020). Among many differences in the FL-HPO problem setting, there are two key points we want to emphasize: (i) Theorem 2.1 presents the first optimality gap in terms of loss function value for the single-shot FL-HPO setting, and can be applied to both algorithmic and model architecture HPs, while existing works either lack theoretical guarantees or establish weaker optimality gap measured by the regret defined for an online setting and are only applicable to single HP optimization. (ii) We only make mild Lipschitz assumption regarding the loss function, and *do not require any assumptions* on the convexity of the loss function, the parties' local data distributions, or the training algorithms, while existing works usually require convexity and certain restrictions on the ML training algorithm to obtain their convergence guarantees.

## 3 EMPIRICAL EVALUATION

In this section, we evaluate FLoRA with different loss surfaces for the FL-HPO problems on a variety of ML models – histograms based gradient boosted (HGB) decision trees (Friedman, 2001), Support Vector Machines (SVM) with RBF kernel and multi-layered perceptrons (MLP), using their respective `scikit-learn` implementations (Pedregosa et al., 2011) on OpenML (Vanschoren et al., 2013) classification problems. First, we fix the number of parties $p = 3$ and compare FLoRA to a baseline on 7 datasets. Then we study the data heterogeneity effect on the performance of FLoRA. Finally, we evaluate FLoRA with different parameter choices, in particular, the number of local HPO rounds and the communication overhead in the aggregation of the per-party (HP, loss) pairs. More comprehensive experimental results and FLoRA performance on real FL systems can be found in Appendix D.

**Baselines.** To appropriately evaluate our proposed single-shot FL-HPO scheme, we need to select a meaningful single-shot baseline. For this, we choose the default HP configuration of `scikit-learn` as the single-shot baseline for two main reasons: (i) the default HP configuration in `scikit-learn` is set manually based on expert prior knowledge and extensive empirical evaluation, and (ii) these are also used as the defaults in the Auto-Sklearn package (Feurer et al., 2015; 2020), one of the leading open-source AutoML python packages, which maintains a carefully selected portfolio of default configurations. While there are some existing schemes for FL-HPO, we are unable to compare FLoRA to them – see Table 1 for detailed comparison.

**Implementation and evaluation metric.** We emulate the final FL (Algorithm 1, line 8) with a centralized training using the pooled data. We chose this implementation because we want to evaluate the final performance of any HP configuration (baseline or recommended by FLoRA) in a statistically robust manner with multiple train/validation splits (for example, via 10-fold cross-validation) instead

of evaluating the performance on a single train/validation. This form of evaluation is extremely expensive to perform in a real FL system and generally not feasible, but allows us to evaluate how the performance of our single-shot HP recommendation fairs against that of the best-possible HP found via a full-scale centralized HPO. In all datasets, we consider the balanced accuracy as the metric we wish to maximize. For the local per-party HPOs (as well as the centralized HPO we execute to compute the regret), we maximize the 10-fold cross-validated balanced accuracy. For Table 3-4, we report the *relative regret*, computed as $\frac{(a^\star - a)}{(a^\star - b)}$, where $a^\star$ is the best metric obtained via the centralized HPO, $b$ is the result of the above single-shot baseline, and $a$ is the result of the HP recommended by FLoRA. The baseline has a relative regret of 1 and smaller values imply better performance. A value larger than 1 implies that the recommended HP performs worse than the single-shot baseline.

**Comparison to single-shot baseline.** We first compare FLoRA with the baseline across different datasets, ML models and FLoRA loss surfaces summarized in Table 3 with the individual results detailed in Appendix D.3. For each method, we report the aggregate performance over all considered datasets in terms of (i) inter-quartile range, (ii) Wins/Ties/Losses of FLoRA w.r.t. the single-shot baseline, and (iii) a one-sided Wilcoxon Signed Ranked Test of statistical significance with the null hypothesis that the median of the difference between the single-shot baseline and FLoRA is positive against the alternative that the difference is negative (implying FLoRA improves over the baseline). Finally, we report an "Overall" performance, further aggregated across all ML models.

All FLoRA loss surfaces show strong performance w.r.t. the single-shot baseline, with significantly more wins than losses, and 3rd-quartile relative regret values less than 1 (indicating improvement over the baseline). All FLoRA loss surfaces have a p-value of less than $0.05$, indicating that we can reject the null hypothesis. Overall, APLM shows the best performance over all loss surfaces, both in terms of Wins/Ties/Losses over the baseline as well as in terms of the Wilcoxon Signed Rank Test, with the highest statistic and a p-value close to $10^{-3}$. APLM also has significantly

Table 4: Effect of increasing the number of parties on FLoRA with different loss surfaces for HGB.

| Data | $p$ | $\gamma_p$ | SGM | SGM+U | MPLM | APLM |
|------|-----|-----------|-----|-------|------|------|
| EEG | 3 | 1.01 | 0.14 | 0.12 | 0.11 | 0.12 |
| 14980 rows | 10 | 1.03 | 0.08 | 0.00 | 0.16 | 0.01 |
| | 25 | 1.08 | 0.35 | 0.92 | 0.17 | 0.04 |
| | 50 | 1.20 | 0.20 | 0.23 | 0.67 | 0.12 |
| Electricity | 3 | 1.01 | 0.17 | 0.14 | 0.09 | 0.12 |
| 45312 rows | 10 | 1.02 | 0.03 | 0.06 | 0.32 | 0.14 |
| | 25 | 1.04 | 0.40 | 0.42 | 1.42 | 0.89 |
| | 50 | 1.07 | 1.57 | 1.57 | 0.89 | 1.13 |
| | 100 | 1.14 | 1.45 | 1.47 | 0.48 | 1.11 |
| Pollen | 3 | 1.02 | 0.43 | 0.54 | 0.43 | 0.69 |
| 3848 rows | 6 | 1.10 | 1.02 | 0.91 | 0.54 | 0.56 |
| | 10 | 1.16 | 1.05 | 0.73 | 0.75 | 1.12 |

lower 3rd-quartile than all other loss surfaces. MPLM appears to have the worst performance but much of that is attributable to a couple of very hard cases with SVM (see Appendix D.3 for detailed discussion). Otherwise, MPLM performs second best both for FL-HPO with HGB and MLP.

**Effect of data heterogeneity.** In the second set of experiments, we study the effect of increasing the number of parties in the FL-HPO problem. For each data set, we increase the number of parties $p$ up until each party has at least 100 training samples. We present the relative regrets in Table 4. It also displays $\gamma_p := \left(1 - \min_{i \in [p]} \mathcal{L}_\star^{(i)}\right)/\left(1 - \max_{i \in [p]} \mathcal{L}_\star^{(i)}\right)$, where $\mathcal{L}_\star^{(i)} = \min_{t \in [T]} \mathcal{L}_t^{(i)}$ is the minimum loss observed during the local HPO at party $i$. This ratio $\gamma_p$ is always greater than 1, and quantifies the inter-party data heterogeneity – precisely $\gamma_p \sim 1 + \widetilde{O}\left(\max_{i,j \in [p]} \mathcal{W}_1(\mathcal{D}_i, \mathcal{D}_j)\right)$ (Appendix C.5).

The results indicate that, with low or moderate increase in $\gamma_p$ (EEG eye state, Electricity for moderate $p$), the proposed scheme is able to achieve low relative regret. However, with significant increase in $\gamma_p$ (Pollen, Electricity with $p = 50, 100$

Table 5: Effect of data hetergeneity on FLoRA with MNIST.

| Method | $\gamma_p$ | SGM | SGM+U | MPLM | APLM |
|--------|-----------|-----|-------|------|------|
| MLP | 1.01 | 0.85 | 0.56 | 0.29 | 0.06 |
| HGB | 1.01 | 0.51 | 0.92 | 0.64 | 0.51 |

and EEG Eye State with $p = 50$), the relative regret increases as well (even $> 1$ in a few cases).

We also simulate a different form of data heterogeneity based on the MNIST dataset and present the results in Table 5. In particular, there are 4 parties in total, with half of the parties having 4 times higher probability to have more even digits while the other half have more odd digits. In most challenging cases, MPLM (the most pessimistic loss function) has the most graceful degradation in relative regret compared to the remaining loss surfaces.

**Communication savings over multi-shot.** Figure 2 presents the communication overhead savings (in terms of the number of FL model trainings) from FLoRA over multi-shot HPO by applying

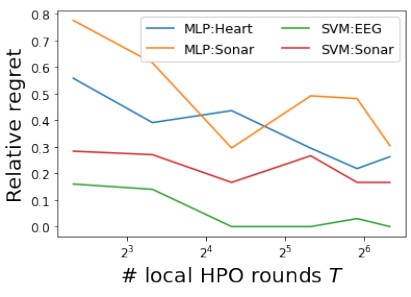
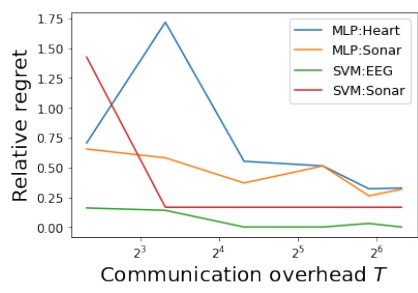

(a) Regret vs # local HPO rounds.

(b) Regret vs # (HP, loss) pairs communicated

Figure 3: Effect of different choices on FLoRA with the APLM loss surface for different methods and datasets. More results and other loss surfaces are presented in Appendix D.6 and D.7.

standard HPO to FL-HPO. For ease of exposition, we present results with HGB with 2 loss surfaces – MPLM and APLM. More results are presented in Appendix D.4. Similar to Figure 1, we report the number of FL model trainings required for multi-shot (■) to match the relative regret achieved by FLoRA (⋆), with the single-shot baseline performance (relative regret of 1 with 1 FL model training, denoted by ●) presented as a reference. In aggregate, FLoRA with APLM achieves a median savings of $8\times$, $15\times$ and $10\times$ over the multi-shot baseline for HGB, SVM and MLP respectively.

**Effect of different choices in FLoRA.** In this set of experiments, we consider FLoRA with the APLM loss surface, and ablate the effect of different choices in FLoRA on 2 datasets each for SVM and MLP. First, we study the impact of the thoroughness of the per-party local HPOs, quantified by the number of HPO rounds $T$ in Figure 3a. The results indicate that for really small $T$ ($< 20$) the relative regret of FLoRA can be very high. However, after that point, the relative regret converges to its best possible value. We present the results for other loss surfaces in Appendix D.6.

We also study the effect of the communication overhead of FLoRA for fixed level of local HPO thoroughness. We assume that each party performs $T = 100$ rounds of local asynchronous HPO. However, instead of sending all $T$ (HP, loss) pairs, we consider sending $T' < T$ of the "best" (HP, loss) pairs – that is, (HP, loss) pairs with the $T'$ lowest losses. Changing the value of $T'$ trades off the communication overhead of the FLoRA step where the aggregators collect the per-party loss pairs (Algorithm 1, line 5). The results for this study are presented in Figure 3b, and indicate that, for really small $T'$, the relative regret can be really high. However, for a moderately high value of $T' < T$, FLoRA converges to its best possible performance. Results and discussions on other loss surfaces are in Appendix D.7.

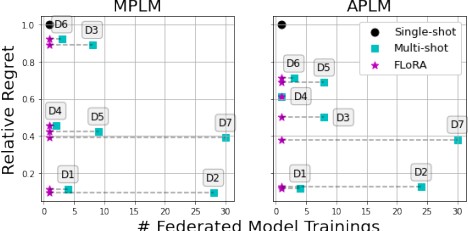

Figure 2: Communication savings of FLoRA compared to "multi-shot" FL-HPO for the same level of relative regret (*lower is better*). Each pair of (⋆, ■) connected by a dashed line corresponds to a dataset labeled as D1-D7 for ease of visualization. See Table 6 in Appendix D.1 for dataset names.

## 4 CONCLUSION AND FUTURE WORK

Effective selection of HPs in FL settings is a challenging problem. In this paper, we introduced FLoRA, a single-shot FL-HPO algorithm that can be applied to any ML model. We provided a theoretical analysis which bounds the optimality gap incurred by the HP selected by FLoRA. Our experiments show that FLoRA can effectively select HPs that outperform the baseline with just a single FL training. As future work, we wish to extend FLoRA to the Combined Algorithm Selection and HPO (CASH) problem (Thornton et al., 2012; Feurer et al., 2015) with or without fixed ML pipeline architecture (Baudart et al., 2021; Hirzel et al., 2022; Katz et al., 2020; Marinescu et al., 2021), especially in the presence of computational and fairness constraints (Liu et al., 2020; Ram et al., 2020), and understand the various theoretical trade-offs in FL-HPO (Ram et al., 2023). One limitation of FLoRA is that it cannot handle HPs that are inactive during local HPO. These include aggregator specific and some FL training specific HPs. It is unlikely that such HPs can be handled in single-shot FL-HPO without any additional information or structure. As future work, we wish to extend FLoRA to handle such HPs in "few-shot" FL-HPO, potentially in conjunction with multi-fidelity HP evaluations.

## REPRODUCIBILITY STATEMENT

The code and instructions to reproduce our numerical results can be found in supplemental materials. For formal definitions, assumptions and proofs of Theorem 2.1, one can find them in Appendix C. We provide a description of the datasets used in our experiments in Appendix D.1. One can also find the original datasets in supplemental materials.

## ACKNOWLEDGEMENTS

We would like to thank Martin Wistuba for his input during the initial discussions on this research thread. We would also like to thank the organizers of the NFFL workshop at NeurIPS'21 for allowing us to present an initial version of this work to a wider audience (Zhou et al., 2021).

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

## A    ADDITIONAL DISCUSSION ON RELATED WORKS

In this section, we provide further discussion of related work and how our work compares against them.

### A.1    MULTI-FIDELITY HPO AND FL-HPO

In centralized HPO, the evaluation of a single HP configuration $\theta$ entails the training of the ML model with the provided HP and training data, the inference of the trained model on some held-out samples (validation data) to get predictions, and the evaluation of the quality of these predictions against available ground-truth. The most computationally expensive part of this process is the model training. Multi-fidelity HPO seeks to reduce the overall computational cost of HPO with cheap evaluations that essentially reduce the model training cost (Swersky et al., 2014; Klein et al., 2017; Li et al., 2018; Falkner et al., 2018). For models trained via some form of (stochastic) gradient descent, a cheap evaluation of a HP configuration is obtained by computing the predictive performance of the model trained with a limited number of gradient descent iterations – the number of iterations is the notion of **budget**. Under the assumption that *more budget does not degrade* the performance of any given HP configuration, multi-fidelity HPO solvers adaptively allocate more budget to more promising HP configurations while discarding low-performing HP configurations cheaply by evaluating them with a low budget.

With models not trained by gradient descent, such as decision-tree based models or nearest-neighbor models or some kernel machines, a commonly used notion of budget is the training set size – cheap evaluations are performed by training the model for any given HP configuration with just a smaller training set size, with the assumption that smaller training set sizes usually speed up the training, which is usually true for most ML models.

One significant distinction between the two notions of budget: With the iteration budget, we are able to progressively train the models for the high performing HP configurations (albeit with additional storage by checkpointing the models after each evaluation with every budget allocation). With the training size budget, we usually have to train the models from scratch for each budget allocation; for example, there is no standard way of progressively updating a decision tree trained on $100$ samples with a new training size allocation of $500$ samples.

### A.1.1    MULTI-FIDELITY IN FL-HPO

In FL-HPO, one of the main bottlenecks is the communication overhead and cheap evaluation in multi-fidelity FL-HPO would require us to control the number of communication rounds in a particular FL training. For gradient descent based training scheme, the number of iterations is a commonly used notion of budget (Khodak et al., 2021) since this number of iterations is a good surrogate for the number of communication rounds that gets allocated to the (cheap) FL training. Hence, multi-fidelity FL-HPO is useful for neural networks.

However, the training set size as a notion of budget (used for other ML models) does not necessarily control the number of rounds of communication in FL-HPO. For example, when training a decision tree of depth 5 in a FL setting (Ong et al., 2020), one would require around 5-6 rounds of communication regardless of the per-party training set sizes – reducing the training set size on each party would not reduce the number of communication rounds in the FL training. Hence, the training set size is not a useful notion of budget for multi-fidelity FL-HPO. For this reason, we focus on the FL-HPO problem where we cannot make use of multi-fidelity HP evaluations. This does not consider the checkpointing overhead present in multi-fidelity FL-HPO with iteration budget.

**Precise communication overheads of different schemes.** We would like to highlight that, even when comparing to some multi-fidelity FL-HPO scheme, the single-shot FLoRA can still provide communications savings. Consider a simplistic setup where each FL training requires $C$ communication rounds. For the single-shot FLoRA, the overall communication overhead will be $O(C)$ since we would need a single FL training and some additional communication which is much much smaller than the FL training communication overhead (see §2.1, Figure 4c and 3b). If multi-shot FL-HPO tries $N$ HP configurations, then the communication overhead will be $O(C \cdot N)$ (see Fig 4b). For a multi-fidelity FL-HPO which leverages some form of successive halving/elimination, it can be shown that the total communication overhead for attempting $N$ HP configurations is $O(C \cdot \log N)$ which is

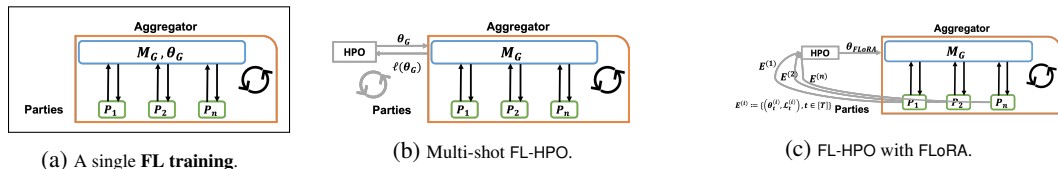

(a) A single **FL training**.    (b) Multi-shot FL-HPO.    (c) FL-HPO with FLoRA.

Figure 4: Visualizations of the communication in FL training and different FL-HPO setup. (4a) A single FL training involves multiple communication rounds. (4b) Multi-shot FL-HPO requires **multiple FL trainings** for each of the different HPs $\boldsymbol{\theta}_G$ evaluated. (4c) FL-HPO with FLoRA requires **a single FL training** for the selected HP $\hat{\boldsymbol{\theta}}^\star$.

significantly smaller than the communication overhead of multi-shot HPO, but still higher than that of FLoRA for moderately high $N$.

## B ADDITIONAL DISCUSSIONS ON LOSS SURFACES

**Single global model (SGM).** We merge all the sets $E = \cup_{i \in [p]} E^{(i)}$ and use it as a training set for a regressor $f : \Theta \to \mathbb{R}$, which considers the HPs $\boldsymbol{\theta} \in \Theta$ as the covariates and the corresponding loss as the dependent variable. For example, we can train a random forest regressor (Breiman, 2001) on this training set $E$. Then we can define the loss surface $\widehat{\ell}(\boldsymbol{\theta}) := f(\boldsymbol{\theta})$. While this loss surface is simple to obtain, it may not be able to handle Non-iid party data distribution well: it is actually overly optimistic – under the assumption that every party generates unique HPs during the local HPO, this single global loss surface would assign a low loss to any HP $\boldsymbol{\theta}$ which has a low loss at any one of the parties. This implies that this loss surface would end up recommending HPs that have low loss in just one of the parties, but not necessarily on all parties.

**Single global model with uncertainty (SGM+U).** Given the merged set $E = \cup_{i \in [p]} E^{(i)}$, we can train a regressor that provides uncertainty quantification around its predictions (such as Gaussian Process Regressor (Williams & Rasmussen, 2006)) as $f : \Theta \to \mathbb{R}, u : \Theta \to \mathbb{R}_+$, where $f(\boldsymbol{\theta})$ is the mean prediction of the model at $\boldsymbol{\theta} \in \Theta$ while $u(\boldsymbol{\theta})$ quantifies the uncertainty around this prediction $f(\boldsymbol{\theta})$. We define the loss surface as $\widehat{\ell}(\boldsymbol{\theta}) := f(\boldsymbol{\theta}) + \alpha \cdot u(\boldsymbol{\theta})$ for some $\alpha > 0$. The uncertainty function $u$ depends on the regressor used to model the loss surface. Gaussian Process Regressors naturally provide uncertainty estimates for predictions. For Random Forest Regressors, uncertainty estimates for predictions have been generated based on the variance of the individual tree predictions in the Random Forest, and we use this in our experiments. For all our experiments, $\alpha = 1$, implies that we are considering a loss value that is a single standard deviation away from the predicted loss.

This loss surface does prefer HPs that have a low loss even in just one of the parties, but it penalizes a HP if the model estimates high uncertainty around this HP. Usually, a high uncertainty around a HP would be either because the training set $E$ does not have many samples around this HP (implying that many parties did not view the region containing this HP as one with low loss), or because there are multiple samples in the region around this HP but parties do not collectively agree that this is a promising region for HPs. Hence this makes SGM+U more desirable than SGM, giving us a loss surface that estimates low loss for HPs that are simultaneously thought to be promising to multiple parties.

**Maximum of per-party local models (MPLM).** We can train one regressor $f^{(i)} : \Theta \to \mathbb{R}, i \in [p]$ with each of the per-party set $E^{(i)}$. Given this, we can construct the loss surface as $\widehat{\ell}(\boldsymbol{\theta}) := \max_{i \in [p]} f^{(i)}(\boldsymbol{\theta})$. This can be seen as a much more pessimistic loss surface, assigning a low loss to a HP only if it has a low loss estimate across all parties.

**Average of per-party local models (APLM).** A less pessimistic version of MPLM would be to construct the loss surface as the average of the per-party regressors $f^{(i)}, i \in [p]$ instead of the maximum, defined as $\widehat{\ell}(\boldsymbol{\theta}) := \frac{1}{p} \sum_{i=1}^{p} f^{(i)}(\boldsymbol{\theta})$. This is also less optimistic than SGM since it will assign a low loss for a HP only if its average across all per-party regressors is low, which implies that all parties observed a relatively low loss around this HP.

Intuitively, we believe that loss surfaces such as SGM+U or APLM would be the most promising while the extremely optimistic and pessimistic SGM and MPLM respectively would be relatively less promising, with MPLM being superior to SGM. But MPLM would also be most robust to data heterogeneity, as evidenced by our empirical evaluations.

## C  THEORETICAL ANALYSIS

In this section, we will provide a detailed and rigorous proof to Theorem 2.1. We first formally define some notation we will use throughout this section in Section C.1, and then we discuss the Lipschitz smoothness assumptions (and possible relaxation) we require to obtain our optimality guarantee in Section C.2. Finally, we establish the sub-optimality of the HP selected by FLoRA in Section C.3 and a byproduct of the result in presented in Section C.4.

### C.1  TECHNICAL DEFINITIONS

**Definition C.1** (Loss functions). For a given set of parties' data $\overline{D} = \{D_i\}_{i \in [p]}$ and any $\boldsymbol{\theta} \in \boldsymbol{\Theta}$, the true target loss (any predictive performance metric, such as, the training loss) can be expressed as:

$$\ell(\boldsymbol{\theta}, \mathcal{D}) := \underbrace{\mathbb{E}_{(x,y) \sim \mathcal{D}}}_{\text{test perf. of trained model}} \mathcal{L}(\underbrace{\mathcal{A}(\boldsymbol{\theta}, \overline{D})}_{\text{trained model}}, (x, y)). \tag{C.1}$$

Here $\mathcal{D}$ is the data distribution of the test set. Let $\tilde{\ell}(\boldsymbol{\theta}, \mathcal{D})$ be an estimate of the loss defined in (C.1) given some validation (holdout) set $D'$ sampled from $\mathcal{D}$, which is the model performance metric during evaluation and/or inference time.

We assume the parties' training sets are collected before the federated learning such that $\overline{D}$ is fixed and unchanged during the HPO and FL processes, in order words, we do not consider streaming data/online setting.

Now we are ready to provide a more general definition of the unified loss surface constructed by FLoRA as follows:

**Definition C.2** (Unified loss surface). Given the local loss surfaces $\widehat{\ell}_i : \boldsymbol{\Theta} \to \mathbb{R}$ for each party $i \in [p]$ generated by $T$ (HP, loss) pairs $\{(\boldsymbol{\theta}_t^{(i)}, \mathcal{L}_t^{(i)})\}_{t \in [T]}$, and the weight vector $\boldsymbol{\alpha} := (\alpha_1, \ldots, \alpha_p)$ where $\alpha_i(\boldsymbol{\theta}) \in [0, 1], \forall i \in [p], \sum_{i=1}^{p} \alpha_i(\boldsymbol{\theta}) = 1$, we can define the global loss surface $\widehat{\ell} : \boldsymbol{\Theta} \to \mathbb{R}$ as

$$\widehat{\ell}(\boldsymbol{\theta}) = \sum_{i=1}^{p} \alpha_i(\boldsymbol{\theta}) \cdot \widehat{\ell}_i(\boldsymbol{\theta}). \tag{C.2}$$

In particular,

    i) If $\alpha_i(\boldsymbol{\theta}) = 1/p, \ \forall i \in [p], \boldsymbol{\theta} \in \boldsymbol{\Theta}$, then this reduces to APLM loss surface.

    ii) If $\alpha_i(\boldsymbol{\theta}) = \mathbb{I}\left(\widehat{\ell}_i(\boldsymbol{\theta}) = \max_{j \in [p]} \widehat{\ell}_j(\boldsymbol{\theta})\right)$, then this reduces to the MPLM loss surface (assuming all $\widehat{\ell}_j(\boldsymbol{\theta})$s are unique).

We formalize the distance metric used in our analysis to evaluate the distance between two given data distributions using 1-Wasserstein distance (Villani, 2021).

**Definition C.3** (1-Wasserstein distance (Villani, 2021)). For two distributions $\mu, \nu$ with bounded support, the 1-Wasserstein distance is defined as

$$\mathcal{W}_1(\mu, \nu) := \sup_{f \in \mathsf{F}_1} \mathbb{E}_{x \sim \mu} f(x) - \mathbb{E}_{x \sim \nu} f(x), \tag{C.3}$$

where $\mathsf{F}_1 = \{f : f \text{ is continuous}, \mathsf{Lipschitz}(f) \leq 1\}$.

We now define a distance metric $d : \boldsymbol{\Theta} \times \boldsymbol{\Theta} \to \mathbb{R}_+$ used in the rest of our analysis. Assuming we have $m$ HPs, if $\boldsymbol{\Theta} \subset \mathbb{R}^m$, then there are various distances available such as $\|\boldsymbol{\theta} - \boldsymbol{\theta}'\|_\rho$ (the $\rho$-norm). The more general case is where we have $R$ continuous/real HPs, $I$ integer HPs, and $C$ categorical HPs; $m = R + I + C$. In that case, $\boldsymbol{\Theta} \subset \mathbb{R}^R \times \mathbb{Z}^I \times \mathbb{C}^C$, and any $\boldsymbol{\theta} = (\boldsymbol{\theta}_\mathbb{R}, \boldsymbol{\theta}_\mathbb{Z}, \boldsymbol{\theta}_\mathbb{C}) \in \boldsymbol{\Theta}_\mathbb{R} \times \boldsymbol{\Theta}_\mathbb{Z} \times \boldsymbol{\Theta}_\mathbb{C}$, where $\boldsymbol{\theta}_\mathbb{R} \in \boldsymbol{\Theta}_\mathbb{R}, \boldsymbol{\theta}_\mathbb{Z} \in \boldsymbol{\Theta}_\mathbb{Z}, \boldsymbol{\theta}_\mathbb{C} \in \boldsymbol{\Theta}_\mathbb{C}$ respectively denote the continuous, integer and categorical HPs

in $\boldsymbol{\theta}$. Distances over $\mathbb{R}^R \times \mathbb{Z}^I$ is available, such as $\rho$-norm. Let $d_{\mathbb{R},\mathbb{Z}} : (\boldsymbol{\Theta}^{\mathbb{R}} \times \boldsymbol{\Theta}_{\mathbb{Z}}) \times (\boldsymbol{\Theta}_{\mathbb{R}} \times \boldsymbol{\Theta}_{\mathbb{Z}}) \to \mathbb{R}_+$ be some such distance. To define distances over categorical spaces, there are some techniques such as one described by Oh et al. (2019):

Assume that each of the $C$ HPs $\boldsymbol{\theta}_{\mathbb{C},k}, k \in [C]$ have $n_k$ categories $\{\xi_{k1}, \xi_{k2}, \ldots, \xi_{kn_k}\}$. Then we define a complete undirected graph $G_k = (V_k, E_k), k \in [C]$ where

- There is a node $N_{kj}$ in $G_k$ for each category $\xi_{kj}$ for each $j \in [n_k]$ and $V_k = \{N_{k1}, \ldots N_{kn_k}\}$.
- There is an undirected edge $(N_{kj}, N_{kj'})$ for each pair $j, j' \in [n_k]$, and $E_k = \{(N_{kj}, N_{kj'}), j, j' \in [n_k]\}$.

Given the per-categorical HP graph $G_k, k \in [C]$, we define the graph Cartesian product $\mathsf{G} = \bigotimes_{k \in [C]} G_k$ and $\mathsf{G} = (\mathsf{V}, \mathsf{E})$ such that

- $\mathsf{V} = \{\mathsf{N}_{(j_1, j_2, \ldots, j_C)} : (\xi_{1j_1}, \xi_{2j_2}, \ldots \xi_{kj_k}, \ldots, \xi_{Cj_C}) \in \boldsymbol{\Theta}_{\mathbb{C}}, j_k \in [n_k] \forall k \in [C]\}$.
- $\mathsf{E} = \{(\mathsf{N}_{(j_1, j_2, \ldots, j_C)}, \mathsf{N}_{(j_1', j_2', \ldots, j_C')}) : \mathsf{IFF} \exists t \in [C] \text{ such that } \forall k \neq t, \xi_{kj_k} = \xi_{kj_k'}, \text{ and } \exists (N_{tj_t}, N_{tj_t'}) \in E_t\}$.

Then for any $\boldsymbol{\theta}_{\mathbb{C}}, \boldsymbol{\theta}_{\mathbb{C}}' \in \boldsymbol{\Theta}_{\mathbb{C}}$ with corresponding nodes $\mathsf{N}, \mathsf{N}' \in \mathsf{V}$, (Oh et al., 2019, Theorem 2.2.1) says that the length of the shortest path between nodes $\mathsf{N}$ and $\mathsf{N}'$ in $\mathsf{G}$ is a distance. We can consider this distance as $d_{\mathbb{C}} : \boldsymbol{\Theta}_{\mathbb{C}} \times \boldsymbol{\Theta}_{\mathbb{C}} \to \mathbb{R}_+$. Of course, there are other ways of defining distances in the categorical space.

Then we can define a distance $d : (\boldsymbol{\Theta}_{\mathbb{R}} \times \boldsymbol{\Theta}_{\mathbb{Z}} \times \boldsymbol{\Theta}_{\mathbb{C}}) \times (\boldsymbol{\Theta}_{\mathbb{R}} \times \boldsymbol{\Theta}_{\mathbb{Z}} \times \boldsymbol{\Theta}_{\mathbb{C}}) \to \mathbb{R}_+$ between two HPs $\boldsymbol{\theta}, \boldsymbol{\theta}'$ as

$$d(\boldsymbol{\theta}, \boldsymbol{\theta}') = d_{\mathbb{R},\mathbb{Z}}((\boldsymbol{\theta}_{\mathbb{R}}, \boldsymbol{\theta}_{\mathbb{Z}}), (\boldsymbol{\theta}_{\mathbb{R}}', \boldsymbol{\theta}_{\mathbb{Z}}')) + d_{\mathbb{C}}(\boldsymbol{\theta}_{\mathbb{C}}, \boldsymbol{\theta}_{\mathbb{C}}'). \tag{C.4}$$

**Proposition C.4.** *Given distance metrics $d_{\mathbb{R},\mathbb{Z}}$ and $d_{\mathbb{C}}$, the function $d : \boldsymbol{\Theta} \times \boldsymbol{\Theta}$ defined in* (C.4) *is a valid distance metric.*

## C.2 LIPSCHITZ CONTINUITY ASSUMPTIONS

To facilitate our analysis later, we make the following Lipschitz smoothness assumptions regarding the loss function $\tilde{\ell}$ and also the per-party loss surface $\widehat{\ell}_i$.

**Assumption C.5** (Lipschitz smoothness)**.** For a fixed data distribution $\mathcal{D}$ and $\forall \boldsymbol{\theta}, \boldsymbol{\theta}' \in \bar{\boldsymbol{\Theta}} \subset \boldsymbol{\Theta}$, we have

$$|\tilde{\ell}(\boldsymbol{\theta}, \mathcal{D}) - \tilde{\ell}(\boldsymbol{\theta}', \mathcal{D})| \leq \tilde{L}(\mathcal{D}) \cdot d(\boldsymbol{\theta}, \boldsymbol{\theta}'), \tag{C.5}$$

$$|\widehat{\ell}_i(\boldsymbol{\theta}) - \widehat{\ell}_i(\boldsymbol{\theta}')| \leq \widehat{L}_i \cdot d(\boldsymbol{\theta}, \boldsymbol{\theta}'), \tag{C.6}$$

where $d(\cdot, \cdot)$ is the distance metric (see (C.4)) defined over the hyper-parameter search space $\boldsymbol{\theta}$. For a fixed set of hyper-parameters $\boldsymbol{\theta} \in \bar{\boldsymbol{\Theta}} \subset \boldsymbol{\Theta}$ and some data distributions $\mathcal{D}$ and $\mathcal{D}'$, we have

$$|\tilde{\ell}(\boldsymbol{\theta}, \mathcal{D}) - \tilde{\ell}(\boldsymbol{\theta}, \mathcal{D}')| \leq \tilde{\beta}(\boldsymbol{\theta}) \cdot \mathcal{W}_1(\mathcal{D}, \mathcal{D}'). \tag{C.7}$$

**Remark.** Note that we explicitly use a $\bar{\boldsymbol{\Theta}} \subset \boldsymbol{\Theta}$ to highlight that we need Lipschitz smoothness only in some particular parts of the HP space. In fact, our analysis only requires the Lipschitz smoothness at $\widehat{\boldsymbol{\theta}}^*$ (HP selected by FLoRA), $\boldsymbol{\theta}^*$ (the optimal HP) and a HP space containing the aforementioned two HPs and the set of HPs tried in local HPO runs, i.e., $\{\boldsymbol{\theta}_t^{(i)}\}_{t \in [T]}$, which most of the time is much smaller than the entire HP search space. Moreover, the above Lipschitz smoothness assumption w.r.t. a general HP space, which could be a combination of continuous and discrete variables, may be strong. We will show later in this section that it can be relaxed to a milder assumption based on the modulus of continuity without significantly affecting our main results. For simplicity, we can always assume that $\tilde{L}(\mathcal{D}) \leq \tilde{L}, \forall \mathcal{D}$ and $\tilde{\beta}(\boldsymbol{\theta}) \leq \tilde{\beta}, \forall \boldsymbol{\theta}$.

For a more general handling, we can consider the notion of modulus of continuity in the form of a increasing real-valued functions $\omega : \mathbb{R}_+ \to \mathbb{R}_+$ with $\lim_{t \to 0} \omega(t) = \omega(0) = 0$. Then we can say that the estimated loss $\tilde{\ell}(\boldsymbol{\theta}, \mathcal{D})$ and the loss surface $\ell_i(\boldsymbol{\theta})$ admits $\tilde{\omega}_{\mathcal{D}}$ and $\widehat{\omega}$ as a modulus of continuity (respectively) if

$$|\ell(\boldsymbol{\theta}, \mathcal{D}) - \ell(\boldsymbol{\theta}', \mathcal{D})| \leq \tilde{\omega}_{\mathcal{D}}(d(\boldsymbol{\theta}, \boldsymbol{\theta}')) \tag{C.8}$$

$$|\widehat{\ell}_i(\boldsymbol{\theta}) - \widehat{\ell}_i(\boldsymbol{\theta}')| \leq \widehat{\omega}(d(\boldsymbol{\theta}, \boldsymbol{\theta}')). \tag{C.9}$$

If we further assume that $\tilde{\omega}_{\mathcal{D}}, \widehat{\omega}$ to be concave, then we can say that these functions are sublinear as follows:

$$\tilde{\omega}_{\mathcal{D}}(t) \leq \tilde{A}_{\mathcal{D}} \cdot t + \tilde{B}_{\mathcal{D}}, \tag{C.10}$$

$$\widehat{\omega}(t) \leq \widehat{A} \cdot t + \widehat{B}. \tag{C.11}$$

These conditions give us (indirectly) something similar in spirit to the guarantees of Lipschitz continuity, but is a more rigorous way of achieving such guarantees.

## C.3   Proof to Theorem 2.1

In this section, we provide detailed proofs to the result we stated in Theorem 2.1 in Section 2.3.

We are interested in providing a bound for the *optimality gap* defined in (2.8), we restate it as following

$$\mathcal{G} := \tilde{\ell}(\widehat{\boldsymbol{\theta}}^\star, \mathcal{D}) - \tilde{\ell}(\boldsymbol{\theta}^\star, \mathcal{D}),$$

where

$$\boldsymbol{\theta}^\star \in \arg\min_{\boldsymbol{\theta} \in \Theta} \tilde{\ell}(\boldsymbol{\theta}, \mathcal{D}). \tag{C.12}$$

Note that this bound is the optimality gap for the output of FLoRA in terms of the estimated loss $\tilde{\ell}$. We (re)state our main results in the following theorem (in a more precise manner than Theorem 2.1).

**Theorem C.6.** *Consider the optimality gap defined in (2.8), where $\widehat{\boldsymbol{\theta}}^\star$ is selected by FLoRA with each party $i \in [p]$ collecting $T$ (HP, loss) pairs $\{(\boldsymbol{\theta}_t^{(i)}, \mathcal{L}_t^{(i)})\}_{t \in [T]}$ during the local HPO run. For a desired data distribution $\mathcal{D} = \sum_{i=1}^{p} w_i \mathcal{D}_i$, where $\{\mathcal{D}_i\}_{i \in [p]}$ are the sets of parties' local data distributions and $w_i \in [0, 1], \forall i \in [p]$, we have*

$$\mathcal{G} \leq 2 \max_{\boldsymbol{\theta} \in \bar{\Theta}} \sum_{i \in [p]} \alpha_i(\boldsymbol{\theta}) \left\{ \tilde{\beta}(\boldsymbol{\theta}) \sum_{j \in [p], j \neq i} w_j \mathcal{W}_1(\mathcal{D}_j, \mathcal{D}_i) + \left( \tilde{L}(\mathcal{D}_i) + \widehat{L}_i \right) \min_{t \in [T]} d(\boldsymbol{\theta}, \boldsymbol{\theta}_t^{(i)}) + \delta_i \right\}, \tag{C.13}$$

*where $\delta_i$ is the maximum per sample training error for the local loss surface $\widehat{\ell}_i$, i.e., $\delta_i = \max_t |\mathcal{L}_t^{(i)} - \widehat{\ell}_i(\boldsymbol{\theta}_t^{(i)})|$. In particular, when all parties have i.i.d. local data distributions, (C.13) reduces to*

$$\mathcal{G} \leq 2 \max_{\boldsymbol{\theta} \in \bar{\Theta}} \sum_{i=1}^{p} \alpha_i(\boldsymbol{\theta}) \left\{ \left( \tilde{L}(\mathcal{D}_i) + \widehat{L}_i \right) \min_{t \in [T]} d(\boldsymbol{\theta}, \boldsymbol{\theta}_t^{(i)}) + \delta_i \right\}.$$

We quantify the relationship between $\widehat{\ell}_i(\boldsymbol{\theta})$ and the estimated loss function $\tilde{\ell}(\boldsymbol{\theta}, \mathcal{D}_i)$ as follows:

$$|\widehat{\ell}_i(\boldsymbol{\theta}) - \tilde{\ell}(\boldsymbol{\theta}, \mathcal{D}_i)| := \epsilon_i(\boldsymbol{\theta}, T). \tag{C.14}$$

**Proposition C.7.** *Consider $\widehat{\boldsymbol{\theta}}^\star$ and $\boldsymbol{\theta}^\star$ are two sets of HP defined in (2.7) and (C.12), respectively, and $\{\mathcal{D}_i\}_{i \in [p]}$ and $\mathcal{D}$ are the sets of parties' local data distributions and the target (global) data distribution we want to learn, for a given HP space such that $\widehat{\boldsymbol{\theta}}^\star, \boldsymbol{\theta}^\star \in \bar{\Theta} \subset \Theta$, we have*

$$\tilde{\ell}(\widehat{\boldsymbol{\theta}}^\star, \mathcal{D}) - \tilde{\ell}(\boldsymbol{\theta}^\star, \mathcal{D}) \leq 2 \max_{\boldsymbol{\theta} \in \bar{\Theta}} \sum_{i \in [p]} \alpha_i(\boldsymbol{\theta}) \left\{ \tilde{\beta}(\boldsymbol{\theta}) \mathcal{W}_1(\mathcal{D}, \mathcal{D}_i) + \epsilon_i(\boldsymbol{\theta}, T) \right\}. \tag{C.15}$$

*Proof.* Consider the definition of $\widehat{\boldsymbol{\theta}}^\star$ and $\boldsymbol{\theta}^\star$, we can obtain

$$\tilde{\ell}(\widehat{\boldsymbol{\theta}}^\star, \mathcal{D}) - \tilde{\ell}(\boldsymbol{\theta}^\star, \mathcal{D}) = \tilde{\ell}(\widehat{\boldsymbol{\theta}}^\star, \mathcal{D}) - \widehat{\ell}(\widehat{\boldsymbol{\theta}}^\star) + \widehat{\ell}(\widehat{\boldsymbol{\theta}}^\star) - \widehat{\ell}(\boldsymbol{\theta}^\star) + \widehat{\ell}(\boldsymbol{\theta}^\star) - \tilde{\ell}(\boldsymbol{\theta}^\star, \mathcal{D})$$

$$\leq 2 \max_{\boldsymbol{\theta} \in \bar{\Theta} \subset \Theta} \left| \tilde{\ell}(\boldsymbol{\theta}, \mathcal{D}) - \widehat{\ell}(\boldsymbol{\theta}) \right|,$$

where the inequality follows from the fact that $\widehat{\ell}(\widehat{\boldsymbol{\theta}}^\star) - \widehat{\ell}(\boldsymbol{\theta}^\star) \leq 0$. Moreover, observe that for any $\boldsymbol{\theta} \in \bar{\boldsymbol{\Theta}} \subset \boldsymbol{\Theta}$, by the definition of $\widehat{\ell}(\boldsymbol{\theta})$ in (C.2), we have

$$
\begin{aligned}
|\tilde{\ell}(\boldsymbol{\theta}, \mathcal{D}) - \widehat{\ell}(\boldsymbol{\theta})| &= \left| \tilde{\ell}(\boldsymbol{\theta}, \mathcal{D}) - \sum_{i \in [p]} \alpha_i(\boldsymbol{\theta}) \cdot \widehat{\ell}_i(\boldsymbol{\theta}) \right| \\
&= \left| \tilde{\ell}(\boldsymbol{\theta}, \mathcal{D}) - \sum_{i \in [p]} \alpha_i(\boldsymbol{\theta}) \cdot \tilde{\ell}(\boldsymbol{\theta}, \mathcal{D}_i) \right. \\
&\quad \left. + \sum_{i \in [p]} \alpha_i(\boldsymbol{\theta}) \cdot \tilde{\ell}(\boldsymbol{\theta}, \mathcal{D}_i) - \sum_{i \in [p]} \alpha_i(\boldsymbol{\theta}) \cdot \widehat{\ell}_i(\boldsymbol{\theta}) \right| \\
&\leq \sum_{i \in [p]} \alpha_i(\boldsymbol{\theta}) \left| \tilde{\ell}(\boldsymbol{\theta}, \mathcal{D}) - \tilde{\ell}(\boldsymbol{\theta}, \mathcal{D}_i) \right| + \sum_{i \in [p]} \alpha_i(\boldsymbol{\theta}) \left| \tilde{\ell}(\boldsymbol{\theta}, \mathcal{D}_i) - \widehat{\ell}_i(\boldsymbol{\theta}) \right| \\
&\leq \sum_{i \in [p]} \alpha_i(\boldsymbol{\theta}) \tilde{\beta}(\boldsymbol{\theta}) \mathcal{W}_1(\mathcal{D}, \mathcal{D}_i) + \sum_{i \in [p]} \alpha_i(\boldsymbol{\theta}) \epsilon_i(\boldsymbol{\theta}, T),
\end{aligned}
$$

where the last inequality follows from assumption (C.7) and definition (C.14). $\qquad \square$

We now dive into each term in (C.15) to provide tight bounds for $\mathcal{W}_1(\mathcal{D}, \mathcal{D}_i)$ and $\epsilon_i(\boldsymbol{\theta}, T)$ in the following propositions.

**Proposition C.8.** *Consider 1-Wasserstein distance we defined in* (C.3), *for a local data distribution $\mathcal{D}_i$ of any party $i$, $i \in [p]$, and $\mathcal{D} = \sum_{i=1}^{p} w_i \mathcal{D}_i$ for some $w_i \in [0, 1], \forall i \in [p]$, we have*

$$
\mathcal{W}_1(\mathcal{D}, \mathcal{D}_i) \leq \sum_{j \in [p], j \neq i} w_j \mathcal{W}_1(\mathcal{D}_j, \mathcal{D}_i). \tag{C.16}
$$

*In particular, when $\mathcal{D}_i$, $i \in [p]$ are i.i.d. data distribution, i.e., all parties in a federated learning system possess i.i.d. local data distribution – that is, $\mathcal{W}_1(\mathcal{D}_j, \mathcal{D}_i) = 0 \forall i, j \in [p]$ – then $\sum_{j \in [p], j \neq i} w_j \mathcal{W}_1(\mathcal{D}_j, \mathcal{D}_i) = 0$. Therefore, $\mathcal{W}_1(\mathcal{D}, \mathcal{D}_i) = 0$, $\forall i \in [p]$.*

*Proof.* By the definition of 1-Wasserstein distance in (C.3) and the fact that $\mathcal{D} = \sum_{i \in [p]} w_i \mathcal{D}_i$, we can obtain

$$
\begin{aligned}
\mathcal{W}_1(\mathcal{D}, \mathcal{D}_i) &= \sup_{f \in \mathsf{F}_1} \mathbb{E}_{(x,y) \sim \mathcal{D}} f(x, y) - \mathbb{E}_{(x_i, y_i) \sim \mathcal{D}_i} f(x_i, y_i) \\
&= \sup_{f \in \mathsf{F}_1} \sum_{j \in [p]} w_j \mathbb{E}_{(x_j, y_j) \sim \mathcal{D}_j} f(x_j, y_j) - \mathbb{E}_{(x_i, y_i) \sim \mathcal{D}_i} f(x_i, y_i) \\
&= \sup_{f \in \mathsf{F}_1} \sum_{i \neq j, j \in [p]} w_j \left( \mathbb{E}_{(x_j, y_j) \sim \mathcal{D}_j} f(x_j, y_j) - \mathbb{E}_{(x_i, y_i) \sim \mathcal{D}_i} f(x_i, y_i) \right) \\
&\leq \sum_{i \neq j, j \in [p]} w_j \left( \sup_{f \in \mathsf{F}_1} \mathbb{E}_{(x_j, y_j) \sim \mathcal{D}_j} f(x_j, y_j) - \mathbb{E}_{(x_i, y_i) \sim \mathcal{D}_i} f(x_i, y_i) \right) \\
&\leq \sum_{i \neq j, j \in [p]} w_j \mathcal{W}_1(\mathcal{D}_j, \mathcal{D}_i).
\end{aligned}
$$

$\qquad \square$

**Proposition C.9.** *For any party $i$, $i \in [p]$, consider a (HP, loss) pair $(\boldsymbol{\theta}_t^{(i)}, \mathcal{L}_t^{(i)})$ collected during the local HPO run for party $i$, for any $\boldsymbol{\theta} \in \bar{\boldsymbol{\Theta}} \subset \boldsymbol{\Theta}$, we have*

$$
\epsilon_i(\boldsymbol{\theta}, T) \leq \left( \tilde{L}(\mathcal{D}_i) + \widehat{L}_i \right) \min_{t \in [T]} d(\boldsymbol{\theta}, \boldsymbol{\theta}_t^{(i)}) + \delta_i, \tag{C.17}
$$

*where $\delta_i = \max_t |\mathcal{L}_t^{(i)} - \widehat{\ell}_i(\boldsymbol{\theta}_t^{(i)})|$ is the maximum per sample training error for the local loss surface $\widehat{\ell}_i$.*

*Proof.* By the definition of $\epsilon_i(\boldsymbol{\theta}, T)$

$$
\begin{aligned}
\epsilon_i(\boldsymbol{\theta}, T) &= \left| \tilde{\ell}(\boldsymbol{\theta}, \mathcal{D}_i) - \widehat{\ell}_i(\boldsymbol{\theta}) \right| \\
&= \left| \underbrace{\tilde{\ell}(\boldsymbol{\theta}, \mathcal{D}_i) - \tilde{\ell}(\boldsymbol{\theta}_t^{(i)}, \mathcal{D}_i)}_{\text{Smoothness of } \tilde{\ell}} + \underbrace{\tilde{\ell}(\boldsymbol{\theta}_t^{(i)}, \mathcal{D}_i) - \widehat{\ell}_i(\boldsymbol{\theta}_t^{(i)})}_{\text{Modeling error}} + \underbrace{\widehat{\ell}_i(\boldsymbol{\theta}_t^{(i)}) - \widehat{\ell}_i(\boldsymbol{\theta})}_{\text{Smoothness of } \widehat{\ell}} \right|,
\end{aligned}
$$

where $\boldsymbol{\theta}_t^{(i)}, t \in [T]$ is *any* one of the HP tried during local HPO run on party $i \in [p]$.

First note that

$$|\tilde{\ell}(\boldsymbol{\theta}_t^{(i)}, \mathcal{D}_i) - \widehat{\ell}_i(\boldsymbol{\theta}_t^{(i)})| = |\mathcal{L}_t^{(i)} - \widehat{\ell}_i(\boldsymbol{\theta}_t^{(i)})|$$
$$\leq \max_t |\mathcal{L}_t^{(i)} - \widehat{\ell}_i(\boldsymbol{\theta}_t^{(i)})| \leq \delta_i.$$

In view of (C.5) and (C.7), we have

$$\epsilon_i(\boldsymbol{\theta}, T) \leq \tilde{L}(\mathcal{D}_i) d(\boldsymbol{\theta}, \boldsymbol{\theta}_t^{(i)}) + \delta_i + \widehat{L}_i d(\boldsymbol{\theta}, \boldsymbol{\theta}_t^{(i)}),$$

which immediately implies the result in (C.17). $\qquad\square$

In view of the results established in Proposition C.7-C.9, we immediately obtain the optimality guarantee presented in Theorem C.6 (also Theorem 2.1).

The following proposition characterizes $\epsilon_i(\boldsymbol{\theta}, T)$ when we relax the Lipschitz continuity assumption with respect to the HP space $\theta$ (required by Proposition C.9) to only modulus continuity defined at the end of Section C.2.

**Proposition C.10.** *Assume that the estimated loss $\tilde{\ell}(\boldsymbol{\theta}, \mathcal{D}_i)$ and the loss surface $\ell_i(\boldsymbol{\theta})$ admit concave functions $\tilde{\omega}_{\mathcal{D}_i}$ and $\widehat{\omega}_i$ respectively as a modulus of continuity with respect to $\boldsymbol{\theta} \in \Theta$ for each party $i \in [p]$. Then, for any party $i$, $i \in [p]$, with the set of (HP, loss) pairs $\{(\boldsymbol{\theta}_t^{(i)}, \mathcal{L}_t^{(i)})\}_{t \in [T]}$ collected during the local HPO run for party $i$, for any $\boldsymbol{\theta} \in \bar{\boldsymbol{\Theta}} \subset \boldsymbol{\Theta}$, there exists $\tilde{A}_{\mathcal{D}_i}, \widehat{A}_i, \tilde{B}_{\mathcal{D}_i}, \widehat{B}_i \geq 0$ such that*

$$\epsilon_i(\boldsymbol{\theta}, T) \leq \left( \tilde{A}_{\mathcal{D}_i} + \widehat{A}_i \right) \min_{t \in [T]} d(\boldsymbol{\theta}, \boldsymbol{\theta}_t^{(i)}) + \tilde{B}_{\mathcal{D}_i} + \widehat{B}_i + \delta_i, \qquad (C.18)$$

*where $\delta_i = \max_t |\mathcal{L}_t^{(i)} - \widehat{\ell}_i(\boldsymbol{\theta}_t^{(i)})|$ is the maximum per sample training error for the local loss surface $\widehat{\ell}_i$.*

*Proof.* By the definition of $\epsilon_i(\boldsymbol{\theta}, T)$

$$\epsilon_i(\boldsymbol{\theta}, T) = \left| \tilde{\ell}(\boldsymbol{\theta}, \mathcal{D}_i) - \widehat{\ell}_i(\boldsymbol{\theta}) \right|$$

$$= \left| \underbrace{\tilde{\ell}(\boldsymbol{\theta}, \mathcal{D}_i) - \tilde{\ell}(\boldsymbol{\theta}_t^{(i)}, \mathcal{D}_i)}_{\text{Smoothness of } \tilde{\ell}} + \underbrace{\tilde{\ell}(\boldsymbol{\theta}_t^{(i)}, \mathcal{D}_i) - \widehat{\ell}_i(\boldsymbol{\theta}_t^{(i)})}_{\text{Modeling error}} + \underbrace{\widehat{\ell}_i(\boldsymbol{\theta}_t^{(i)}) - \widehat{\ell}_i(\boldsymbol{\theta})}_{\text{Smoothness of } \widehat{\ell}} \right|,$$

where $\boldsymbol{\theta}_t^{(i)}, t \in [T]$ is *any* one of the HP tried during local HPO run on party $i \in [p]$.

First note that

$$|\tilde{\ell}(\boldsymbol{\theta}_t^{(i)}, \mathcal{D}_i) - \widehat{\ell}_i(\boldsymbol{\theta}_t^{(i)})| = |\mathcal{L}_t^{(i)} - \widehat{\ell}_i(\boldsymbol{\theta}_t^{(i)})|$$
$$\leq \max_t |\mathcal{L}_t^{(i)} - \widehat{\ell}_i(\boldsymbol{\theta}_t^{(i)})|$$
$$\leq \delta_i.$$

In view of (C.8) and (C.9), we have

$$\epsilon_i(\boldsymbol{\theta}, T) \leq \tilde{\omega}_{\mathcal{D}_i}(d(\boldsymbol{\theta}, \boldsymbol{\theta}_t^{(i)})) + \delta_i + \widehat{\omega}(d(\boldsymbol{\theta}, \boldsymbol{\theta}_t^{(i)})),$$
$$\leq \delta_i + \min_{t \in [T]} \left( \tilde{\omega}_{\mathcal{D}_i}(d(\boldsymbol{\theta}, \boldsymbol{\theta}_t^{(i)})) + \widehat{\omega}(d(\boldsymbol{\theta}, \boldsymbol{\theta}_t^{(i)})) \right).$$

Concavity of a function $\omega : [0, \infty] \to [0, \infty]$ implies that there exists $A, B > 0$ such that $\omega(t) \leq At + B$. Using that, we can find some $\tilde{A}_{\mathcal{D}_i}, \widehat{A}_i, \tilde{B}_{\mathcal{D}_i}, \widehat{B}_i > 0$ which allows us to simplify the above to

$$\epsilon_i(\boldsymbol{\theta}, T) \leq \delta_i + (\tilde{A}_{\mathcal{D}_i} + \widehat{A}) \cdot \min_{t \in [T]} d(\boldsymbol{\theta}, \boldsymbol{\theta}_t^{(i)}) + (\tilde{B}_{\mathcal{D}_i} + \widehat{B}).$$

$\qquad\square$

## C.4 RELATIVE REGRETS

As a byproduct, we can also provide a bound for the following relative regret we use in our experiments.

**Corollary C.11.** *Let us assume $\widehat{\boldsymbol{\theta}}^{\star}$ and $\boldsymbol{\theta}^{\star}$ are defined in (2.7) and (C.12), and $\bar{\boldsymbol{\theta}}^{\star}$ and $\boldsymbol{\theta}_b$ are the hyper-parameter settings selected by centralized HPO and some baseline hyper-parameters, respectively, then we can bound the relative regret as follows, for a given data distribution $\mathcal{D}$, we have*

$$
\frac{\tilde{\ell}(\bar{\boldsymbol{\theta}}^{\star}, \mathcal{D}) - \tilde{\ell}(\widehat{\boldsymbol{\theta}}^{\star}, \mathcal{D})}{\tilde{\ell}(\bar{\boldsymbol{\theta}}^{\star}, \mathcal{D}) - \tilde{\ell}(\boldsymbol{\theta}_b, \mathcal{D})}
$$

$$
\leq \frac{2 \max_{\boldsymbol{\theta} \in \bar{\Theta}} \sum_{i=1}^{p} \alpha_i(\boldsymbol{\theta}) \left\{ \tilde{\beta}(\boldsymbol{\theta}) \sum_{j \in [p], j \neq i} w_j \mathcal{W}_1(\mathcal{D}_j, \mathcal{D}_i) + \left( \tilde{L}(\mathcal{D}_i) + \widehat{L}_i \right) \min_{t \in [T]} d(\boldsymbol{\theta}, \boldsymbol{\theta}_t^{(i)}) + \delta_i \right\}}{\widehat{\ell}(\boldsymbol{\theta}_b, \mathcal{D}) - \widehat{\ell}(\bar{\boldsymbol{\theta}}^{\star}, \mathcal{D})}.
$$

$$(C.19)$$

*Proof.* By the definition of relative regret, we have

$$
\frac{\tilde{\ell}(\bar{\boldsymbol{\theta}}^{\star}, \mathcal{D}) - \tilde{\ell}(\widehat{\boldsymbol{\theta}}^{\star}, \mathcal{D})}{\tilde{\ell}(\bar{\boldsymbol{\theta}}^{\star}, \mathcal{D}) - \tilde{\ell}(\boldsymbol{\theta}_b, \mathcal{D})} = \frac{\tilde{\ell}(\widehat{\boldsymbol{\theta}}^{\star}, \mathcal{D}) - \tilde{\ell}(\bar{\boldsymbol{\theta}}^{\star}, \mathcal{D})}{\tilde{\ell}(\boldsymbol{\theta}_b . \mathcal{D}) - \tilde{\ell}(\bar{\boldsymbol{\theta}}^{\star}, \mathcal{D})}
$$

$$
\leq \frac{\tilde{\ell}(\widehat{\boldsymbol{\theta}}^{\star}, \mathcal{D}) - \tilde{\ell}(\boldsymbol{\theta}^{\star}, \mathcal{D})}{\widehat{\ell}(\boldsymbol{\theta}_b, \mathcal{D}) - \widehat{\ell}(\bar{\boldsymbol{\theta}}^{\star}, \mathcal{D})},
$$

where the last inequality follows from the fact that $\boldsymbol{\theta}^{\star}$ is the minimizer of $\tilde{\ell}(\boldsymbol{\theta}, \mathcal{D})$. Moreover, in view of the result in Theorem C.6, the result in (C.19) follows. □

## C.5 QUANTIFYING DATA HETEROGENEITY IN FL-HPO

As defined in §3, we utilize $\gamma_p$ as a surrogate to quantify the heterogeneity between the per-party data distributions. Here we will motivate this choice. By definition

$$
\gamma_p = \frac{1 - \min_{i \in [p]} \min_{t \in [T]} \mathcal{L}_t^{(i)}}{1 - \max_{i \in [p]} \min_{t \in [T]} \mathcal{L}_t^{(i)}}
$$

$$
= \frac{1 - \min_{t \in [T]} \mathcal{L}_t^{(\widetilde{i})}}{1 - \min_{t \in [T]} \mathcal{L}_t^{(\widehat{i})}} \quad \text{where} \quad \widetilde{i} := \arg \min_{i \in [p]} \min_{t \in [T]} \mathcal{L}_t^{(i)} \quad \text{and} \quad \widehat{i} := \arg \max_{i \in [p]} \min_{t \in [T]} \mathcal{L}_t^{(i)}
$$

$$
= \frac{1 - \min_{t \in [T]} \mathcal{L}_t^{(\widehat{i})} + \min_{t \in [T]} \mathcal{L}_t^{(\widehat{i})} - \min_{t \in [T]} \mathcal{L}_t^{(\widetilde{i})}}{1 - \min_{t \in [T]} \mathcal{L}_t^{(\widehat{i})}}
$$

$$
= 1 + \frac{\min_{t \in [T]} \mathcal{L}_t^{(\widehat{i})} - \min_{t \in [T]} \mathcal{L}_t^{(\widetilde{i})}}{1 - \min_{t \in [T]} \mathcal{L}_t^{(\widehat{i})}}
$$

$$
\approx 1 + \frac{\tilde{\ell}(\boldsymbol{\theta}_{\star}^{(\widehat{i})}, \mathcal{D}_{\widehat{i}}) - \tilde{\ell}(\boldsymbol{\theta}_{\star}^{(\widetilde{i})}, \mathcal{D}_{\widetilde{i}})}{1 - \min_{t \in [T]} \mathcal{L}_t^{(\widehat{i})}} \quad \text{where } \boldsymbol{\theta}_{\star}^{(\widetilde{i})}, \boldsymbol{\theta}_{\star}^{(\widehat{i})} \text{ are the best HPs seen at parties } \widetilde{i}, \widehat{i} \text{ resp.}
$$

$$
= 1 + \frac{\tilde{\ell}(\boldsymbol{\theta}_{\star}^{(\widehat{i})}, \mathcal{D}_{\widehat{i}}) - \ell(\boldsymbol{\theta}_{\star}^{(\widetilde{i})}, \mathcal{D}_{\widehat{i}}) + \ell(\boldsymbol{\theta}_{\star}^{(\widetilde{i})}, \mathcal{D}_{\widehat{i}}) - \tilde{\ell}(\boldsymbol{\theta}_{\star}^{(\widetilde{i})}, \mathcal{D}_{\widetilde{i}})}{1 - \min_{t \in [T]} \mathcal{L}_t^{(\widehat{i})}}
$$

$$
\leq 1 + \frac{\tilde{\beta} \cdot \mathcal{W}_1(\mathcal{D}_{\widetilde{i}}, \mathcal{D}_{\widehat{i}})}{1 - \min_{t \in [T]} \mathcal{L}_t^{(\widehat{i})}} \leq 1 + \frac{\tilde{\beta} \cdot \max_{i,j \in [p]} \mathcal{W}_1(\mathcal{D}_i, \mathcal{D}_j)}{1 - \min_{t \in [T]} \mathcal{L}_t^{(\widehat{i})}} \approx 1 + \mathcal{O}\left( \max_{i,j \in [p]} \mathcal{W}_1(\mathcal{D}_i, \mathcal{D}_j) \right).
$$

This implies that $\gamma_p$ is closely tied to the maximum 1-Wassertein distance between any pair of per-party distributions.

# D    EXPERIMENTAL SETTING

## D.1    DATASET DETAILS

For our evaluation of single-shot HPO, we consider 7 binary classification datasets of varying sizes and characteristics from OpenML (Vanschoren et al., 2013) such that there is at least a significant room for improvement over the single-shot baseline performance. We consider datasets which have at least $> 3\%$ potential improvement in balanced accuracy for gradient boosted decision trees. Note that this only ensures room for improvement for HGB, while highlighting cases with no room for improvement for SVM and MLP as we see in our results.

The details of the binary classification datasets used in our evaluation is reported in Table 6. We report the 10-fold cross-validated balanced accuracy of the default HP configuration on each of datasets with centralized training. The "Gap" column for the results for all datasets and models in §D.3 denote the difference between the best 10-fold cross-validated balanced accuracy obtained via centralized HPO and the 10-fold cross-validated balanced accuracy of the default HP configuration.

Table 6: OpenML binary classification dataset details

| Index | Data | rows | columns | class sizes |
|---|---|---|---|---|
| D1 | EEG eye state | 14980 | 14 | (8257, 6723) |
| D2 | Electricity | 45312 | 8 | (26075, 19237) |
| D3 | Heart statlog | 270 | 13 | (150, 120) |
| D4 | Oil spill | 937 | 49 | (896, 41) |
| D5 | Pollen | 3848 | 5 | (1924, 1924) |
| D6 | Sonar | 208 | 61 | (111, 97) |
| D7 | PC3 | 1563 | 37 | (1403, 160) |

## D.2    SEARCH SPACE

We use the search space definition used in the NeurIPS 2020 Black-box optimization challenge (https://bbochallenge.com/), described in details in the API documentation[1].

### D.2.1    HISTOGRAM BASED GRADIENT BOOSTED TREES

Given this format for defining the HPO search space, we utilize the following precise search space for the `HistGradientBoostingClassifier` in `scikit-learn`:

```
api_config = {
    "max_iter": {"type": "int", "space": "linear", "range": (10, 200)},
    "learning_rate": {"type": "real", "space": "log", "range": (1e-3, 1.0)},
    "min_samples_leaf": {"type": "int", "space": "linear", "range": (1, 40)},
    "l2_regularization": {"type": "real", "space": "log", "range": (1e-4, 1.0)},
}
```

The HP configuration we consider for the single-shot baseline described in §3 is as follows:

```
config = {
    "max_iter": 100,
    "learning_rate": 0.1,
    "min_samples_leaf": 20,
    "l2_regularization": 0,
}
```

### D.2.2    KERNEL SVM WITH RBF KERNEL

For `SVC(kernel="rbf")` in `scikit-learn`, we use the following search space:

---

[1]`https://github.com/rdturnermtl/bbo_challenge_starter_kit/`
`#configuration-space`

```
api_config = {
    "C": {"type": "real", "space": "log", "range": (0.01, 1000.0)},
    "gamma": {"type": "real", "space": "log", "range": (1e-5, 10.0)},
    "tol": {"type": "real", "space": "log", "range": (1e-5, 1e-1)},
}
```

The single shot baseline we consider for `SVC` from Auto-sklearn (Feurer et al., 2015) is:

```
config = {
    "C": 1.0,
    "gamma": 0.1,
    "tol": 1e-3,
```

### D.2.3 MULTI-LAYERED PERCEPTRONS

For the `MLPClassifier(solver="adam")` from `scikit-learn`, we consider both architectural HP such as `hidden-layer-sizes` as well as optimizer parameters such as `alpha` and `learning-rate-init` for the Adam optimizer (Kingma & Ba, 2015). We consider the following search space:

```
api_config = {
    "hidden_layer_sizes": {"type": "int", "space": "linear", "range": (50, 200)},
    "alpha": {"type": "real", "space": "log", "range": (1e-5, 1e1)},
    "learning_rate_init": {"type": "real", "space": "log", "range": (1e-5, 1e-1)},
}
```

We utilize the following single shot baseline:

```
config = {
    "hidden_layer_sizes: 100,
    "alpha": 1e-4,
    "learning_rate_init": 1e-3,
}
```

We fix the remaining HPs of `MLPClassifier` as with values used by Auto-sklearn.

```
activation="relu",
early_stopping=True,
shuffle=True,
batch_size="auto",
tol=1e-4,
validation_fraction=0.1,
beta_1=0.9,
beta_2=0.999,
epsilon=1e-8,
```

### D.3 DETAILED RESULTS OF COMPARISON AGAINST BASELINES

Here we present the relevant details and the performance of FLoRA on the FL-HPO of (i) histograms based gradient boosted trees (HGB) in Table 7), (ii) nonlinear support vector machines (SVM) in Table 8, and (iii) multi-layered perceptrons (MLP) in Table 9. We use the search spaces and the single-shot baselines presented in §D.2. We utilize all 7 datasets for each of the method *except* for the Electricity dataset with SVM because of the infeasible amount of time taken by SVM on this dataset. For each setup, we report the following:

- Performance of the single-shot baseline ("SSBaseline"),
- the best centralized HPO performance ("Best"),
- the available "Gap" for improvement,
- the minimum accuracy of the best local HP across all parties "PMin" := $\min_{i \in [p]} \max_t (1 - \tilde{\ell}(\boldsymbol{\theta}_t^{(i)}, \mathcal{D}_i))$

- the maximum accuracy of the best local HP across all parties "PMax" $:= \max_{i \in [p]} \max_t (1 - \tilde{\ell}(\boldsymbol{\theta}_t^{(i)}, \mathcal{D}_i))$
- $\gamma_p = {}^{\text{PMax}}/_{\text{PMin}}$, and finally
- the regret for each of the considered loss surfaces in FLoRA.

For each of the three methods, we also report the aggregate performance over all considered datasets in terms of mean ± standard deviation ("mean±std"), inter-quartile range ("IQR"), Wins/Ties/Losses of FLoRA with respect to the single-shot baseline ("W/T/L"), and a one-sided Wilcoxon Signed Ranked Test of statistical significance ("WSRT") with the null hypothesis that the median of the difference between the single-shot baseline and FLoRA is positive against the alternative that the difference is negative (implying FLoRA improves over the baseline).' These aggregate metrics are collected in Table 10 along with a set of final aggregate metrics across all datasets and methods.

Table 7: HGB

| Data | SSBaseline | Best | Gap | PMin | PMax |
|------|-----------|------|-----|------|------|
| PC3 | 58.99 | 63.81 | 4.82 | 61.67 | 64.37 |
| Pollen | 48.86 | 52.21 | 3.35 | 51.83 | 52.64 |
| Electricity | 87.75 | 92.84 | 5.10 | 88.42 | 89.19 |
| Sonar | 87.43 | 91.25 | 3.82 | 83.75 | 88.33 |
| Heart Statlog | 79.42 | 85.58 | 6.17 | 78.00 | 86.50 |
| Oil Spill | 63.22 | 74.58 | 11.36 | 68.16 | 82.16 |
| EEG Eye State | 89.96 | 94.66 | 4.70 | 91.80 | 92.29 |

| Data | $\gamma_p$ | SGM | SGM+U | MPLM | APLM |
|------|-----------|-----|-------|------|------|
| PC3 | 1.04 | 0.66 | 0.72 | 0.39 | 0.38 |
| Pollen | 1.02 | 0.43 | 0.54 | 0.43 | 0.69 |
| Electricity | 1.01 | 0.17 | 0.14 | 0.09 | 0.12 |
| Sonar | 1.05 | 1.33 | 0.41 | 0.92 | 0.71 |
| Heart Statlog | 1.11 | 0.69 | 0.55 | 0.89 | 0.50 |
| Oil Spill | 1.21 | 0.47 | 1.13 | 0.46 | 0.61 |
| EEG Eye State | 1.01 | 0.14 | 0.12 | 0.11 | 0.12 |
| mean±std | | $0.56 \pm 0.37$ | $0.52 \pm 0.32$ | $0.47 \pm 0.31$ | $0.45 \pm 0.23$ |
| IQR | | [0.30, 0.47, 0.68] | [0.27, 0.54, 0.64] | [0.25, 0.43, 0.67] | [0.25, 0.50, 0.65] |
| WTL | | 6/0/1 | 6/0/1 | 7/0/0 | 7/0/0 |
| WSRT | | (26, 0.02126) | (27, 0.01400) | (28, 0.00898) | (28, 0.00898) |

**HGB.** The results in Table 7 indicate that, in almost all cases, with all loss functions, FLoRA is able to improve upon the baseline to varying degrees (there is only one case where SGM performs worse than the baseline on Sonar). On average (across the datasets), SGM+U, MPLM and APLM perform better than SGM as we expected. MPLM performs better than SGM both in terms of average and standard deviation. Looking at the individual datasets, we see that, for datasets with low $\gamma_p$ (EEG eye state, Electricity), all the proposed loss surface have low relative regret, indicating that the problem is easier as expected. For datasets with high $\gamma_p$ (Heart statlog, Oil spill), the relative regret of all loss surfaces are higher (but still much smaller than 1), indicating that our proposed single-shot scheme can show improvement even in cases where there is significant difference in the per-party losses (and hence datasets).

**SVM.** For SVM we continue with the datasets selected using HGB (datasets with a "Gap" of at least 3%). Of the 7 datasets (Table 6), we skip Electricity because it takes a prohibitively long time for SVM to be trained on this dataset with a single HP. So we consider 6 datasets in this evaluation and present the corresponding results in Table 8. Of the 6, note that 2 of these datasets (Pollen, Heart Statlog) have really small "Gap" (highlighted in red in Table 8). Moreover, 2 of the datasets (Heart statlog, Oil Spill) also have really high $\gamma_p$ indicating a high level of heterogeneity between the per-party distributions (again highlighted in red). In this case, there are a couple of datasets (Oil Spill and Pollen) where FLoRA is unable to show any improvement over the single-shot baseline (see underlined entries in Table 8), but both these cases either have a small or moderate "Gap" and/or have a high $\gamma_p$. Moreover, in one case, MPLM incurs a regret of 6.8, but this is a case with really high $\gamma_p = 1.14$ – MPLM rejects any HP that has a low score in even one of the parties, and in that process

Table 8: SVM

| Data | SSBaseline | Best | Gap | PMin | PMax |
|---|---|---|---|---|---|
| Pollen | 49.48 | 50.30 | 0.82 | 51.55 | 53.55 |
| Sonar | 80.20 | 89.29 | 9.09 | 83.33 | 87.92 |
| Heart Statlog | 83.67 | 84.92 | 1.25 | 77.00 | 88.00 |
| Oil Spill | 82.76 | 86.54 | 3.78 | 77.14 | 88.45 |
| EEG Eye State | 50.24 | 60.51 | 10.28 | 69.54 | 71.72 |
| PC3 | 74.03 | 77.96 | 3.92 | 75.26 | 76.95 |

| Data | $\gamma_p$ | SGM | SGM+U | MPLM | APLM |
|---|---|---|---|---|---|
| Pollen | 1.04 | 1.35 | 1.45 | 2.84 | 2.30 |
| Sonar | 1.06 | 0.17 | 0.17 | 0.27 | 0.17 |
| Heart Statlog | 1.14 | 0.00 | 0.00 | 6.80 | 0.67 |
| Oil Spill | 1.15 | 1.28 | 1.16 | 1.12 | 0.41 |
| EEG Eye State | 1.03 | -0.01 | -0.01 | -0.02 | -0.01 |
| PC3 | 1.02 | 0.59 | 0.79 | 0.70 | 0.79 |
| mean±std | | 0.56 ± 0.57 | 0.59 ± 0.58 | 1.95 ± 2.35 | 0.72 ± 0.76 |
| IQR | | [0.04, 0.38, 1.11] | [0.04, 0.48, 1.07] | [0.38, 0.91, 2.41] | [0.23, 0.54, 0.76] |
| WTL | | 4/0/2 | 4/0/2 | 3/0/3 | 5/0/1 |
| WSRT | | (18, 0.05793) | (17, 0.08648) | (9, 0.62342) | (15, 0.17272) |

reject all promising HPs since the local HPOs on these disparate distributions did not concentrate on the same region of the HP space, thereby incuring a high MPLM loss in almost all regions of the HP where some local HPO focused on. Other than these expected hard cases, FLoRA is able to improve upon the baseline in most cases, and achieve optimal performance (zero regret) in a few cases (EEG Eye State, Heart Statlog).

Table 9: MLP-Adam

| Data | SSBaseline | Best | Gap | PMin | PMax |
|---|---|---|---|---|---|
| Pollen | 50.39 | 51.26 | 0.87 | 51.46 | 52.23 |
| Electricity | 76.95 | 78.06 | 1.11 | 77.01 | 77.39 |
| Sonar | 61.63 | 79.32 | 17.69 | 69.17 | 78.75 |
| Heart Statlog | 72.17 | 85.17 | 13.00 | 79.50 | 89.50 |
| Oil Spill | 50.00 | 65.22 | 15.22 | 54.83 | 63.63 |
| EEG Eye State | 49.99 | 51.66 | 1.67 | 50.02 | 51.84 |
| PC3 | 50.00 | 59.56 | 9.56 | 53.47 | 56.60 |

| Data | $\gamma_p$ | SGM | SGM+U | MPLM | APLM |
|---|---|---|---|---|---|
| Pollen | 1.02 | 1.88 | 1.45 | 1.45 | 1.31 |
| Electricity | 1.00 | 0.24 | 0.41 | 0.16 | 0.53 |
| Sonar | 1.14 | 0.26 | 0.55 | 0.52 | 0.39 |
| Heart Statlog | 1.13 | 0.46 | 0.37 | 0.42 | 0.28 |
| Oil Spill | 1.16 | 0.80 | 1.03 | 1.00 | 0.79 |
| EEG Eye State | 1.04 | 0.99 | 0.99 | 0.99 | 0.99 |
| PC3 | 1.06 | 0.96 | 1.00 | 0.89 | 0.90 |
| mean±std | | 0.80 ± 0.53 | 0.83 ± 0.37 | 0.78 ± 0.40 | 0.74 ± 0.34 |
| IQR | | [0.36, 0.80, 0.97] | [0.48, 0.99, 1.01] | [0.47, 0.89, 1.00] | [0.46, 0.79, 0.95] |
| WTL | | 6/0/1 | 4/1/2 | 5/1/1 | 6/0/1 |
| WSRT | | (21, 0.11836) | (15, 0.17272) | (18, 0.05793) | (24, 0.04548) |

**MLP.** We consider all 7 datasets for the evaluation of FLoRA on FL-HPO for MLP HPs and present the results in Table 9. As with SVM, there are a few datasets with a small room for improvement ("Gap") and/or high $\gamma_p$, again highlighted in red in Table 9. In some of these cases, FLoRA is unable to improve upon the single-shot baseline (Pollen, EEG Eye State). Other than these hard cases, FLoRA again able to show significant improvement over the single-shot baseline, with APLM performing the best.

**Aggregate.** The results for all the methods and datasets are aggregated in Table 10. All FLoRA loss surfaces show strong performance with respect to the single-shot baseline, with significantly more wins than losses, and 3rd-quartile regret values less than 1 (indicating improvement over the baseline). All FLoRA loss surfaces have a p-value of less than $0.05$, indicating that we can reject the null hypothesis. Overall, APLM shows the best performance over all loss surfaces, both in terms of Wins/Ties/Losses over the baseline as well as in terms of the Wilcoxon Signed Rank Test, with the highest statistic and a p-value close to $10^{-3}$. APLM also has significantly lower 3rd-quartile than all other loss surfaces. MPLM appears to have the worst performance but much of that is attributable to the really high regret of 6.8 and 2.84 it received for SVM with Heart Statlog and Pollen (both hard cases as discussed earlier). Otherwise, MPLM performs second best both for FL-HPO with HGB and MLP.

### D.4 COMMUNICATION SAVINGS OF FLoRA OVER MULTI-SHOT FL-HPO

Here we compare the communication savings one would obtain from FLoRA relative to a multi-shot FL-HPO scheme. FLoRA is a single-shot FL-HPO scheme and hence utilizes a single federated model training (Algorithm 1, line 8). In contrast to a single-shot solution, we can try to solve the FL-HPO problem (2.5) in a manner that allows multiple federated model trainings. In that case, we could utilize existing HPO schemes such a HyperOpt (Bergstra et al., 2011) for FL-HPO, where each HP proposal is evaluated with a complete federated model training. While this multi-shot FL-HPO scheme would have a very high communication overhead, we expect this scheme to find HPs and corresponding models that have better predictive performance than single-shot schemes. In fact, given enough number of federated model trainings, we expect this multi-shot FL-HPO scheme to achieve 0 relative regret.

With such a baseline, we perform a comparison with FLoRA where we note the number of federated model trainings required by the multi-shot FL-HPO to reach the predictive performance (in terms of the relative regret) of FLoRA. Since each federated training incurs a large amount of communication overhead, a larger number of federated model trainings imply a larger communication overhead. This number of federated model training serves as a surrogate for the communication gain of FLoRA over the multi-shot baseline to achieve the same level of performance. We present these results for FL-HPO of HGB (Figure 5), SVM (Figure 6) and MLP (Figure 7). In each of these figures, we present the communication savings for all the four different loss surfaces, and each pair of ($\star$, $\blacksquare$) connected by a dashed line corresponds to a single dataset label from D1-D7. The names (and details) of the dataset corresponding to these dataset labels is provided in Table 6.

The results for HGB in Figure 5 indicate that, in most datasets, across all loss surfaces, FLoRA can provide almost $10\times$ savings and upto $20-30\times$ in a few cases. Overall, SGM performs the worst with

Table 10: Aggregate Table

| Agg. | Method | SGM | SGM+U | MPLM | APLM |
|---|---|---|---|---|---|
| mean $\pm$ std. | HGB | $0.56 \pm 0.37$ | $0.52 \pm 0.32$ | $0.47 \pm 0.31$ | $0.45 \pm 0.23$ |
| | SVM | $0.56 \pm 0.57$ | $0.59 \pm 0.58$ | $1.95 \pm 2.35$ | $0.72 \pm 0.76$ |
| | MLP | $0.80 \pm 0.53$ | $0.83 \pm 0.37$ | $0.78 \pm 0.40$ | $0.74 \pm 0.34$ |
| | Overall | $\mathbf{0.64 \pm 0.51}$ | $\mathbf{0.64 \pm 0.51}$ | $1.02 \pm 1.46$ | $\mathbf{0.63 \pm 0.50}$ |
| IQR | HGB | [0.30, 0.47, 0.68] | [0.27, 0.54, 0.64] | [0.25, 0.43, 0.67] | [0.25, 0.50, 0.65] |
| | SVM | [0.04, 0.38, 1.11] | [0.04, 0.48, 1.07] | [0.38, 0.91, 2.41] | [0.23, 0.54, 0.76] |
| | MLP | [0.36, 0.80, 0.97] | [0.48, 0.99, 1.01] | [0.47, 0.89, 1.00] | [0.46, 0.79, 0.95] |
| | Overall | [**0.22**, **0.53**, 0.97] | [0.32, 0.55, 1.01] | [0.36, 0.61, 0.99] | [0.36, 0.57, **0.79**] |
| W/T/L | HGB | 6/0/1 | 6/0/1 | 7/0/0 | 7/0/0 |
| | SVM | 4/0/2 | 4/0/2 | 3/0/3 | 5/0/1 |
| | MLP | 6/0/1 | 4/1/2 | 5/1/1 | 6/0/1 |
| | Overall | 16/0/4 | 14/1/5 | 15/1/4 | **18/0/2** |
| WSRT 1 sided | HGB | (26, 0.02126) | (27, 0.01400) | (28, 0.00898) | (28, 0.00898) |
| | SVM | (18, 0.05793) | (17, 0.08648) | (9, 0.62342) | (15, 0.17272) |
| | MLP | (21, 0.11836) | (15, 0.17272) | (18, 0.05793) | (24, 0.04548) |
| | Overall | (174, 0.00499) | (164, 0.00272) | (141, 0.03206) | **(183.5, 0.00169)** |

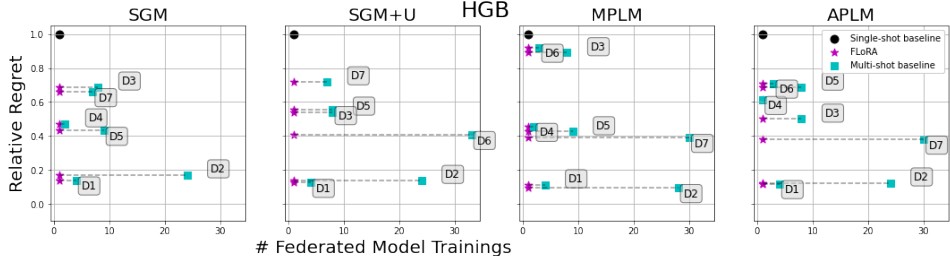

Figure 5: Communication overhead of multi-shot HPO compared to FLoRA to match the relative regret of FLoRA for FL-HPO with Histogram based Gradient Boosted Decision Trees (HGB). The dataset indices are based on the Table 6.

a inter-quartile range (IQR) of $3.5 - 8.5\times$ savings. SGM+U improves over it to provide a savings IQR of $5.5 - 16\times$, while MPLM and APLM provide a savings IQR of $3.5 - 18.5\times$ and $3.5 - 16\times$ respectively.

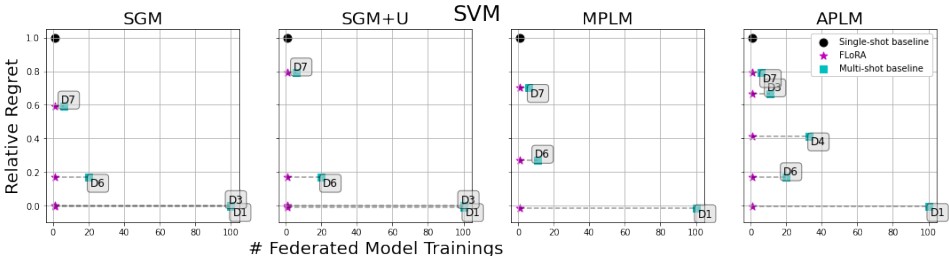

Figure 6: Communication overhead of multi-shot HPO compared to FLoRA to match the relative regret of FLoRA for FL-HPO with Support Vector Machines (SVM). The dataset indices are based on the Table 6.

The results for SVM in Figure 6 indicate that the communication savings of FLoRA are much more spread out. Note that, as mentioned in Appendix D.3, the Electricity dataset (D2) is not considered in the SVM FL-HPO since it takes prohibitively long to execute the centralized HPO. Furthermore, we have also excluded the datasets in each plot where the relative regret for FLoRA is greater than 1 (see Table 8) since in that case, FLoRA is not even able to improve upon the single-shot baseline. There are more cases in this set of results where the savings from FLoRA are less than $10\times$, but there are also cases where FLoRA provides close to $100\times$ savings. Overall, APLM provides the most robust performance in terms of savings with a savings IQR of $7.25 - 29.75\times$, while MPLM provides the worst performance with a savings IQR of $1.75 - 9.75\times$. SGM and SGM+U provide a similar wide-ranging savings IQR of $2.25 - 80\times$.

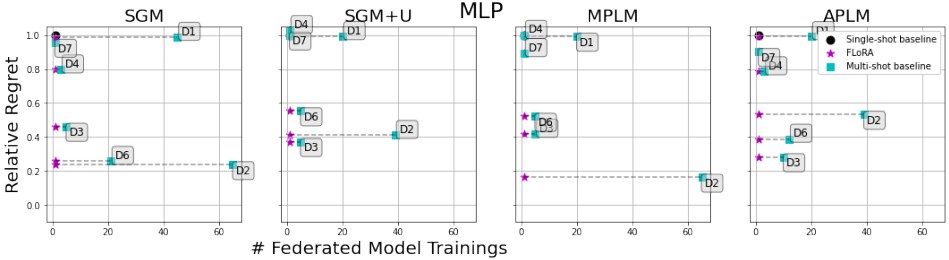

Figure 7: Communication overhead of multi-shot HPO compared to FLoRA to match the relative regret of FLoRA for FL-HPO with Multi-layered Perceptrons (MLP). The dataset indices are based on the Table 6.

Table 11: Effect of increasing the number of parties on FLoRA with all 4 loss surfaces for HGB.

| Data | $p$ | $\gamma_p$ | SGM | SGM+U | MPLM | APLM |
|---|---|---|---|---|---|---|
| EEG Eye State | 3 | 1.01 | 0.14 | 0.12 | 0.11 | 0.12 |
| 14980 samples | 6 | 1.01 | 0.07 | 0.00 | 0.07 | 0.09 |
| | 10 | 1.03 | 0.08 | 0.00 | 0.16 | 0.01 |
| | 25 | 1.08 | 0.35 | 0.92 | 0.17 | 0.04 |
| | 50 | 1.20 | 0.20 | 0.23 | 0.67 | 0.12 |
| Electricity | 3 | 1.01 | 0.17 | 0.14 | 0.09 | 0.12 |
| 45312 samples | 6 | 1.01 | 0.25 | 0.21 | 0.18 | 0.13 |
| | 10 | 1.02 | 0.03 | 0.06 | 0.32 | 0.14 |
| | 25 | 1.04 | 0.40 | 0.42 | 1.42 | 0.89 |
| | 50 | 1.07 | 1.57 | 1.57 | 0.89 | 1.13 |
| | 100 | 1.14 | 1.45 | 1.47 | 0.48 | 1.11 |
| Pollen | 3 | 1.02 | 0.43 | 0.54 | 0.43 | 0.69 |
| 3848 samples | 6 | 1.10 | 1.02 | 0.91 | 0.54 | 0.56 |
| | 10 | 1.16 | 1.05 | 0.73 | 0.75 | 1.12 |

The results for MLP are presented in Figure 7. In this case, the communication savings from FLoRA again range from less than $10\times$ to over $90\times$. APLM again provides the most robust performance with a savings IQR of $5.5 - 16\times$. SGM+U and MPLM have the same savings IQR of $3 - 13\times$. SGM has the most wide-ranging savings IQR of $2 - 33\times$.

Overall, the results across all loss surfaces, datasets and ML models indicate that FLoRA with any loss surface leads to a significant communication savings over multi-shot FL-HPO. Among the loss surfaces, APLM provides one of the best savings among the loss surfaces, while being the most robust across ML models and datasets. SGM+U also provides better savings than SGM. MPLM has the least favorable savings because of its pessimism. However, as shown in other experiments, MPLM is able to provide strong performance relative to the remaining loss surfaces especially when the FL-HPO problem is especially challenging (as with the case of increasing number of parties discussed in Appendix D.5 or the non-IID-ness of the per-party data distribution highlighted in Appendix D.8).

## D.5 EFFECT OF INCREASING THE NUMBER OF PARTIES

We now study the effect of increasing the number of parties in the FL-HPO problem on training HGB model on 3 datasets . For each data set, we increase the number of parties $p$ up until each party has at least 100 training samples. We present the relative regrets for all the loss surfaces in Table 11 (A complete version of Table 4 in Section 3). It also displays $\gamma_p := \left(1 - \min_{i \in [p]} \mathcal{L}_\star^{(i)}\right) / \left(1 - \max_{i \in [p]} \mathcal{L}_\star^{(i)}\right)$, where $\mathcal{L}_\star^{(i)} = \min_{t \in [T]} \mathcal{L}_t^{(i)}$ is the minimum loss observed during the local asynchronous HPO at party $i$. This ratio $\gamma_p$ is always greater than 1, and highlights the difference in the observed performances across the parties. A ratio closer to 1 indicates that all the parties have relatively similar performances on their respective training data, while a ratio much higher than 1 indicating significant discrepancy between the per-party performances, implicitly indicating the difference in the per-party data distributions.

We notice that increasing the number of parties does not have a significant effect on $\gamma_p$ for the Electricity dataset until $p = 100$, but significantly increases for the Pollen dataset earlier (making the problem harder). For the EEG eye state, the increase in $\gamma_p$ with increasing $p$ is moderate until $p = 50$. The results indicate that, with low or moderate increase in $\gamma_p$ (EEG eye state, Electricity for moderate $p$), the proposed scheme is able to achieve low relative regret – the increase in the number of parties does not directly imply degradation in performance. However, with significant increase in $\gamma_p$ (Pollen, Electricity with $p = 50, 100$ and EEG Eye State with $p = 50$), we see a significant increase in the relative regret (eventually going over 1 in a few cases). In this challenging case, MPLM (the most pessimistic loss function) has the most graceful degradation in relative regret compared to the remaining loss surfaces.

Table 12: Effect of $T$.

| Method | data | $T$ | $\gamma_p$ | SGM | SGM+U | MPLM | APLM |
|---|---|---|---|---|---|---|---|
| MLP | Heart Statlog | 5 | 1.13 | 0.58 | 0.33 | 0.22 | 0.56 |
| | | 10 | 1.13 | 0.33 | 0.16 | 0.60 | 0.39 |
| | | 20 | 1.13 | 0.49 | 0.15 | 0.24 | 0.44 |
| | | 40 | 1.13 | 0.44 | 0.30 | 0.42 | 0.29 |
| | | 60 | 1.13 | 0.37 | 0.15 | 0.33 | 0.22 |
| | | 80 | 1.13 | 0.35 | 0.40 | 0.35 | 0.26 |
| MLP | Sonar | 5 | 1.14 | 0.38 | 0.38 | 0.51 | 0.78 |
| | | 10 | 1.14 | 0.45 | 0.23 | 0.43 | 0.62 |
| | | 20 | 1.14 | 0.39 | 0.24 | 0.36 | 0.30 |
| | | 40 | 1.14 | 0.23 | 0.37 | 0.65 | 0.49 |
| | | 60 | 1.14 | 0.53 | 0.14 | 0.34 | 0.48 |
| | | 80 | 1.14 | 0.46 | 0.07 | 0.19 | 0.30 |
| SVM | Sonar | 5 | 1.06 | 0.17 | 0.17 | 1.16 | 0.28 |
| | | 10 | 1.06 | 0.17 | 0.43 | 0.34 | 0.27 |
| | | 20 | 1.06 | 0.17 | 0.17 | 0.17 | 0.17 |
| | | 40 | 1.06 | 0.17 | 0.17 | 0.22 | 0.27 |
| | | 60 | 1.06 | 0.17 | 0.17 | 0.17 | 0.17 |
| | | 80 | 1.06 | 0.17 | 0.11 | 0.27 | 0.17 |
| SVM | EEG | 5 | 1.03 | -0.01 | -0.01 | 0.92 | 0.16 |
| | | 10 | 1.03 | -0.01 | -0.01 | -0.01 | 0.14 |
| | | 20 | 1.03 | -0.01 | -0.01 | -0.00 | -0.00 |
| | | 40 | 1.03 | -0.02 | -0.02 | 0.01 | 0.00 |
| | | 60 | 1.03 | -0.01 | -0.01 | 0.02 | 0.03 |
| | | 80 | 1.03 | -0.01 | -0.01 | 0.01 | -0.0' |

## D.6 EFFECT OF THE NUMBER OF LOCAL HPO ROUNDS PER PARTY

In this experiment, we report additional results to study the effect of the "thoroughness" of the local HPO runs (in terms of the number of HPO rounds $T$) on the overall performance of FLoRA for all the loss surfaces in Table 12. In almost all cases, FLoRA does not require $T$ to be too large to get enough information about the local HPO loss surface to get to its best possible performance.

## D.7 EFFECT OF COMMUNICATION OVERHEAD

While in the previous experiment, we studied the effect of the thoroughness of the local HPO runs on the performance of FLoRA, here we consider a subtly different setup. We assume that each party performs $T = 100$ rounds of local asynchronous HPO. However, instead of sending all $T$ (HP, loss) pairs, we consider sending $T' < T$ of the "best" (HP, loss) pairs – that is, (HP, loss) pairs with the $T'$ lowest losses. Changing the value of $T'$ trades off the communication overhead of the FLoRA step where the aggregators collect the per-party loss pairs (Algorithm 1, line 5). We consider 2 datasets each for 2 of the methods (SVM, MLP) and all the loss surfaces for FLoRA, and report all the results in Table 13.

## D.8 EFFECT OF NON-IID PARTY DATA DISTRIBUTION

In this section, we simulate different degree of non-iidness for parties' local data distribution based on MNIST dataset[2]. In particular, for "imb1" there are 4 parties in total, half of the parties (with 4 times more probability to) have more even digits while the other half have more odd digits. In "imb2" case, parties' local data distributions are more heterogeneous, as there are 4 parties in the FL system with first 3 parties having 2 distinguish class labels and the last parties have the rest 4 class labels, i.e., each party only accesses a unique subset of class labels. For both simulated imbalanced datasets, all parties have 500 training images and 2500 testing images. We use again balanced accuracy as the target performance metric, and relative regret as the metric to measure the performance of FLoRA.

---

[2]We download MNIST dataset from http://yann.lecun.com/exdb/mnist/

Table 13: Effect of the number of best (HP, loss) pairs $T' < T$ sent to aggregator by each party after doing local HPO with $T = 100$.

| Method | data | $T' < T$ | $\gamma_p$ | SGM | SGM+U | MPLM | APLM |
|--------|------|------|------|------|------|------|------|
| MLP | Heart Statlog | 5 | 1.13 | 0.33 | 0.27 | 0.38 | 0.71 |
| | | 10 | 1.13 | 0.35 | 0.31 | 0.33 | 1.72 |
| | | 20 | 1.13 | 0.42 | 0.39 | 2.02 | 0.55 |
| | | 40 | 1.13 | 0.34 | 0.44 | 0.88 | 0.51 |
| | | 60 | 1.13 | 0.38 | 0.22 | 0.31 | 0.32 |
| | | 80 | 1.13 | 0.34 | 0.38 | 0.22 | 0.33 |
| MLP | Sonar | 5 | 1.14 | 0.39 | 0.50 | 1.78 | 0.65 |
| | | 10 | 1.14 | 0.73 | 0.18 | 1.66 | 0.58 |
| | | 20 | 1.14 | 0.20 | 0.41 | 1.23 | 0.37 |
| | | 40 | 1.14 | 0.60 | 0.42 | 0.18 | 0.51 |
| | | 60 | 1.14 | 0.10 | 0.33 | 0.55 | 0.26 |
| | | 80 | 1.14 | 0.47 | 0.41 | 0.34 | 0.32 |
| SVM | EEG Eye State | 5 | 1.03 | -0.02 | -0.01 | 0.39 | 1.02 |
| | | 10 | 1.03 | -0.01 | -0.01 | 1.02 | 1.02 |
| | | 20 | 1.03 | -0.01 | -0.01 | 0.01 | -0.01 |
| | | 40 | 1.03 | -0.01 | -0.01 | -0.00 | -0.01 |
| | | 60 | 1.03 | -0.01 | -0.01 | -0.01 | -0.01 |
| | | 80 | 1.03 | -0.01 | -0.01 | -0.01 | -0.01 |
| SVM | Sonar | 5 | 1.06 | 0.17 | 0.17 | 0.43 | 1.43 |
| | | 10 | 1.06 | 0.17 | 0.17 | 0.17 | 0.17 |
| | | 20 | 1.06 | 0.17 | 0.17 | 0.22 | 0.17 |
| | | 40 | 1.06 | 0.17 | 0.38 | 0.27 | 0.17 |
| | | 60 | 1.06 | 0.17 | 0.43 | 0.27 | 0.17 |
| | | 80 | 1.06 | 0.17 | 0.43 | 0.27 | 0.17 |

Table 14: Effect of Non-iidness among parties' local data distribution.

| Method | data | $\gamma_p$ | SGM | SGM+U | MPLM | APLM |
|--------|------|------|------|------|------|------|
| MLP | MNIST-imb1 | 1.01 | 0.85 | 0.56 | 0.29 | 0.06 |
| MLP | MNIST-imb2 | 1.03 | 20.63 | 1.53 | 0.71 | 1.10 |
| SVM | MNIST-imb1 | 1.01 | 0.01 | 0.02 | 0.01 | 0.02 |
| HGB | MNIST-imb1 | 1.01 | 0.51 | 0.92 | 0.64 | 0.51 |

We summarize the results in Table 14. First of all, FLoRA can improve over the baseline HPs in all imbalance scenarios for both MLP and SVM models. Secondly, we can see that overall MPLM (the most pessimistic loss surface) is the most robust loss surface which can always obtain improvements over the baseline HPs for all scenarios. More specifically, when the degree of non-iidness is mild ("imb1" case), all loss surfaces can improve over the baseline HPs. In "imb2" case when the degree of non-iidness is more intense, we can see that the performance of FLoRA with SGM loss surface drops significantly. This is also supported by our intuitions and theoretical analysis.

## D.9 FEDERATED LEARNING TESTBED EVALUATION

We now conduct experiments for histrogram boosted tree model in a FL testbed, utilizing IBM FL library (Ludwig et al., 2020; Ong et al., 2020), More specifically, we reserved $40\%$ of oil spill and electricity and $20\%$ of EEG eye state as global hold-out set only for evaluating the final FL model performance. Each party randomly sampled from the rest of the original dataset to obtain their own training dataset. We use the same HP search space as in Appendix D.2. We report the balanced accuracy of any HP (baseline or recommended by FLoRA) on a single train/test split. Given balanced accuracy as the evaluation metric, we utilize (1 - balanced accuracy) as the loss $\mathcal{L}_t^{(i)}$ in Algorithm 1 Each party will run HPO to generate $T = 500$ (HP, loss) pairs and use those pairs to generate loss surface either collaboratively or by their own according to different aggregation procedures described

Table 15: Performance of FLoRA with the IBM-FL system in terms of the *balanced accuracy* on a holdout test set (higher is better). The baseline is still the default HP configuration of `HistGradientBoostingClassifier` in `scikit-learn`.

| Data | # parties | # training data per party | Baseline | SGM | SGM+U | MPLM | APLM |
|------|-----------|---------------------------|----------|-----|-------|------|------|
| Oil spill | 3 | 200 | 0.5895 | **0.7374** | 0.5909 | 0.7061 | 0.7332 |
| EEG eye state | 3 | 3,000 | 0.8864 | 0.9153 | 0.9211 | **0.9251** | 0.9245 |
| Electricity | 6 | 4,000 | 0.8448 | 0.8562 | **0.8627** | 0.8621 | 0.8624 |

in §2.2. Once the loss surface is generated, the aggregator uses Hyperopt (Bergstra et al., 2011) to select the best HP candidate and train a federated XGBoost model via the IBM FL library using the selected HPs. Table 15 summarizes the experimental results for 3 datasets, indicating that FLoRA can significantly improve over the baseline in IBM FL testbed.

