# OpenReview forum: "Single-shot General Hyper-parameter Optimization for Federated Learning"
_ICLR.cc/2023/Conference — ICLR 2023 notable top 25%_

### Official Review · Reviewer_mUdQ · 2022-10-24

**Confidence:** 2
**Clarity, Quality, Novelty And Reproducibility:** The quality, clarity and originality …
**Correctness:** 3
**Technical Novelty And Significance:** 2
**Empirical Novelty And Significance:** 3
**Recommendation:** 6

**Strength And Weaknesses:**

Pros:
1. The studies problem is very important because hyperparameter tuning for federated learning under some privacy considerations and distributed nature turns out to be not trivial. The theory is a good plus with milder assumptions and higher generality.

2. It is a general method that can be used in any ML model and the tabular data can also be trained, which is a great expansion in FL-HPO. The number of communication rounds is small since it utilizes a single-shot way that all the (HP, loss) pairs are transferred at a time.

3. The article is logical and most parts are clearly explained. The proposed algorithm (combined with some popular solvers like Bayesian Optimization) is reasonable and seems to work well given the results in the experiments.

Cons:

1. Some parts can be more clearer. For example, on page 6, what is the uncertainty function $u(\theta)$, and how to choose the regularization parameter $\alpha$ ?

2. One key step in my opinion is the regression step to find the mapping f from HP to loss, i.e., line 6. This provides a possibility to reduce the communication rounds because the form of the regressor f is explicitly obtained. However, how does this regression step performs? Is it a linear regression or more complicated processes? How about the computational overhead of this process? I may miss something here, but I wish a clarification here.

3. Figure 4 is a good visualization of the idea, which can be put in a more conspicuous way so that readers can understand it more rapidly.

4. The authors mention that the considered setting has an asynchronous HPOS, but it seems to me that Federated Learning often follows a synchronized nature. Can the authors elaborate this a little bit?

**Summary Of The Paper:**

This paper focuses on hyper-parameter optimization for the federated learning (FL- HPO) problem which is important because the choice of the HPs can have a dramatic impact on the performance. However, there are still some challenges in this area, including the unavailability of the whole dataset, high communication costs, and no FL-HPO approach dealing with a non-neural network. In this case, the author put forward a general algorithm called FLoRA which can train both neural networks and non-neural networks in a single-shot way so that the number of communication rounds is highly saved. The single-shot method is to use different HPs in each client and calculates the loss, forming (HP, loss) pairs. Then, the center collects all (HP, loss) pairs from all clients and generates a loss surface using these data so that the best model can be selected. Moreover, how to generate loss surfaces is also discussed in detail, including SGM, SGM+U, APLM, and MPLM whose performances are also compared in the experiment.

**Summary Of The Review:**

Overall, this paper provides a neat method with good explanations and some theoretical justifications. The experiments seem to support the proposed method. I tend to weakly accept this paper given some concerns above and my low confidence score. However, I am happy to adjust my scores based on other reviewers’ comments and the authors’ response.

---

> ### Author Response · Authors · 2022-11-11
> **Thank you for the review!**
>
> We thank the reviewer for spending time reviewing our work and providing constructive feedback to us. We are happy to see that the reviewer recognizes the importance of the studied problem and the generality of the solution we have proposed, highlighting the single-shot nature, and the applicability to any ML model and tabular data.
> We address the reviewer's comments and concerns as follows:
>
> ---------------------------------
>
> * **Function $u$ and selection of $\alpha$ for SGM+U**
> > Some parts can be more clearer. For example, on page 6, what is the uncertainty function $u(\theta)$, and how to choose the regularization parameter $\alpha$?
>
> ---------------------------------
>
> The uncertainty function depends on the regressor used to model the loss surface. Gaussian Process Regressors naturally provide uncertainty estimates for predictions. For Random Forest Regressors, uncertainty estimates for predictions have been generated based on the variance of the individual tree predictions in the Random Forest, and we use this in our experiments.
>
> For all our experiments, $\alpha = 1$, implies that we are considering a loss value that is a single standard deviation away from the predicted loss.
>
> ---------------------------------
>
> * **More information on Regressor**
> > One key step in my opinion is the regression step to find the mapping f from HP to loss, i.e., line 6. This provides a possibility to reduce the communication rounds because the form of the regressor f is explicitly obtained. However, how does this regression step performs? Is it a linear regression or more complicated processes? How about the computational overhead of this process? I may miss something here, but I wish a clarification here.
>
> ---------------------------------
>
> FLoRA can work with any regular machine learning regressors, such as linear regressor, Gaussian Process regressor, and random forest regressor, etc. As we have discussed in the paragraph after Table 3 on Page 7, "if we use non-parametric regression models as the loss surfaces the per-sample training error can be arbitrarily small". In our experiments, we used random forest as the regressor to obtain loss surfaces since they naturally handle continuous and discrete HPs.
>
> Regarding the computational overhead to obtain the regressor, the training of the loss surface incurs a small amount of computational cost either at the aggregator side or at the party side. However, compared to the local training of regular federated learning in the cross-silo settings where each party may have a huge amount of data, the additional computation overheads for obtaining loss surfaces is very limited, since these loss surfaces can be simple regressor models and only trained on a small amount of data points (e.g., hundreds of (HP, loss) pairs).
> The computational cost is equivalent to training a model with around 100 samples, each with less than 10 features, and hence is extremely fast.
>
> One can also balance the performance of FLoRA and the computation and communication overheads incurred by FLoRA, see our ablation experiments in Fig. 3.
>
> ---------------------------------
>
> * **Figure 4**
> > Figure 4 is a good visualization of the idea, which can be put in a more conspicuous way so that readers can understand it more rapidly.
> ---------------------------------
>
> We thank the reviewer for the suggestion and consolidate Figure 4 and Table 1 in a way that best positions our proposed scheme against existing literature.
>
> ---------------------------------
>
> * **Asynchronous HPOs**
> > The authors mention that the considered setting has an asynchronous HPOS, but it seems to me that Federated Learning often follows a synchronized nature. Can the authors elaborate this a little bit?
>
> ---------------------------------
>
> In usual FL, a single model training is performed in a synchronous way. And with FLoRA, once the final HP has been selected, we train the final model in a synchronous manner.
>
> The local HPO process in FLoRA is performed in an asynchronous manner (i.e., run independently by each party), mostly (i) to keep the overall rounds of communication as small as possible, and (ii) to accommodate for the heterogeneous resource availability and data sizes on the different parties, each requiring different amounts of time for the local HPO; coordinating after each local HPO round is more challenging than coordinating once at the end of the local HPO.
>
> This restriction can be removed and we can perform more coordinated local HPOs if the FL setup allows for it.
>
> ---------------------------------

---

### Official Review · Reviewer_4yTc · 2022-10-24

**Confidence:** 4
**Correctness:** 3
**Technical Novelty And Significance:** 2
**Empirical Novelty And Significance:** 3
**Recommendation:** 6

**Clarity, Quality, Novelty And Reproducibility:**

Clarify: the paper is in general clearly written. The proposed single-shot solution is well motivated.

Quality: Since the paper is more empirical than theoretical, the quality of the empirical evaluations has room for improvement, for which I've listed some points under "Weakness" above.

Novelty: The proposed method is intuitive and is in fact not so novel. But the method is well motivated and reasonable.

Reproducibility: The code is uploaded for reproducibility.

**Strength And Weaknesses:**

Strengths:
- The studied problem, i.e., hyperparameter tuning in the federated setting, is highly relevant.
- The single-shot nature of the proposed method is particularly desirable, since previous methods usually require a lot of additional communication rounds.
- Some of the theoretical insights in Section 2.3 are very interesting, such as those which show the dependence of the optimality gap on the data heterogeneity and the quality of the local HP optimizations.

Weaknesses:
- The proposed method does not allow tuning per-party local HPs (paragraph above equation 2.5), I think this may be a limitation of the proposed method, since due to the heterogeneity of the agents, the best HPs for different agents are highly likely to be different.
- The proposed algorithm (Algorithm 1) requires every agent to pass the raw HP observations to the central server, but wouldn't this lead to concerns regarding privacy? An important principle in FL is that the raw data of an agent can never be passed, and previous papers (see paper [1] below) have pointed out that the hyperparameters of an ML model can indeed contain sensitive information.
[1] Differentially private Bayesian optimization, ICML 2015.
- In the experiments, the proposed method is only compared with a very weak baseline which simply uses the default HP from the package (it is not even optimized by any algorithm). I understand that most of the previous works on FL-HPO may not be comparable, but I think the proposed algorithm should have been compared with more competitive baselines. For example, a natural baseline I can think of is that after collecting all HP observations from all agents, you can simply return the HP that gives the lowest loss (among all HP observations from all agents). I think this makes sense because the loss surface aggregation is the most important technical contribution of the paper, hence this baseline which simply removes this component can be used to evaluate whether the proposed loss surface aggregation is actually useful. Furthermore, when comparing the "communication savings over multi-shot" (top paragraph of page 9), can you compare with some of the previous works on FL-HPO (i.e., those mentioned in Section 1.1) which are multi-shot methods?
- Another important weakness of the empirical evaluation is that the FL algorithm is in fact emulated using centralized training (mentioned at the bottom of page 7). I think this makes the empirical evaluations unrealistic, and may not be able to reflect some unique features/challenges of the FL setting such as heterogeneity.
- Table 4: some of the results are in fact worse than the baseline (which is in fact not so competitive as I discussed above) especially when the number of agents is large, which is worrying for the practical deployment of the proposed method especially in problems with a large number of agents.
- Top paragraph of page 6: how exactly is the uncertainty quantification function u designed? I guess it's supposed to be larger if an HP has been explored by a smaller number of agents?
- Top paragraph of page 7: it is claimed that the theoretical results imply that "the optimality gap depends only on the HP trials $\theta^{(i)}_t$ that are closest to the optimal HP setting". However, I think this may not be accurate. This is because there is a max over all $\theta\in\mathcal{\Theta}$ on the right hand side of equation 2.9, therefore, to make the term $\min\_{t\in[T]}d(\theta,\theta^{(i)\_t})$ small for all $\theta$, we need to make sure that the HPs $\theta^{(i)}\_t$'s are uniformly distributed in the entire domain. Please correct me if my understanding is incorrect here.
- How exactly does the "Multi-shot baseline" in Figure 1 and Figure 2 work? This was never explained.
- This may not be a weakness but rather a suggestion. The local HP optimization step for every agent (line 3 of Algorithm 1) may have room for improvement. Since now we have the opportunity to let every agent independently run HP optimization, we can in fact coordinate the local HP optimizations such that they can combine to lead to more accurate global loss surface estimation. For example, you can consider the method of "distributed exploration" from the work of Dai et al. (2021), and let every agent explore only a small local region of the entire search space.
- [minor] I think it will be very helpful if you could explain in the introduction and abstract that being "single-shot" means only requiring a single-round of communication. It may be misleading to those not from the field.
- [minor] Section 1.1, second paragraph: in "Federated Network Architecture Search", I think Network should be Neural.

**Summary Of The Paper:**

The paper proposes a method for tuning the hyperparameters of ML models in the federated setting which only requires a single round of communication. The proposed method is general-purpose since it can be applied to tune the hyperparameters of any ML model, not only neural networks as in some of the previous works. The proposed method firstly lets every agent run local hyperparameter optimization, and then collect all hyperparameters from all agents to fit a loss function, which is then minimized to select the final hyperparameters.

**Summary Of The Review:**

The method of this paper is well motivated, but I have many major concerns regarding the method and the empirical evaluations. I've listed my concerns under "Weaknesses" above, the first 5 points under "Weaknesses" are my most important concerns.

---

> ### Author Response · Authors · 2022-11-11
> **Thank you for the review! Response Thread (1/5)**
>
> We thank the reviewer for spending time reviewing our work and providing thorough constructive feedback to us. We appreciate the reviewer for highlighting the relevance of the studied problem, and the advantages of the proposed single-shot scheme and its analysis. We address the reviewer's comments and concerns as follows:
>
>
> ---------------------------------
>
> * **Limitation on per-party local HPs**
> > The proposed method does not allow tuning per-party local HPs (paragraph above equation 2.5), I think this may be a limitation of the proposed method, since due to the heterogeneity of the agents, the best HPs for different agents are highly likely to be different.
>
>
> ---------------------------------
>
> This is indeed a limitation of FLoRA. However, we want to point out that heterogeneity (in the per-party data distributions) does not necessarily lead to parties requiring different local HPs. In fact, some established FL algorithms, such as FedProx [Li et al 2020] and Scaffold [Karimireddy et al 2020] do not suggest parties to use different local HPs.
>
> As we have candidly stated in the paragraph above (2.5) and also in our  Conclusion (Section 4), the single-shot nature of FLoRA allows us to save significant communication overheads for performing HPO for FL comparing to other multi-shot FL-HPO approaches, however, one of the limitations of FLoRA is that it can not handle personalized federated learning.
> In this paper, we do not focus on parties' local HPs, whereas we focus on global HPs, such as the number of trees, the number of layers in the neural network, and the global learning rate, etc. Note that although focusing on tuning global HPs, FLoRA is **the first single-shot approach to tune global HPs** (requiring a single FL training). To the best of our knowledge, all existing works need **multi-shot** (multiple FL trainings) to tune global HPs.
>
> We accept that we do not focus on some hyperparameters that may be involved in certain FL algorithms, but we do not think that our contributions are limited since we open up the possibility of tuning the algorithm and architecture-related hyperparameters of any ML method in a FL setting with a single FL training, and FLoRA is agnostic to model types.
>
> ---------------------------------
>
> * **Privacy concerns**
> > The proposed algorithm (Algorithm 1) requires every agent to pass the raw HP observations to the central server, but wouldn't this lead to concerns regarding privacy? ...
>
> ---------------------------------
>
> The reviewer has raised a very interesting point on privacy guarantees provided by FL-HPO approaches.
> We first want to point out that our proposed solution FLoRA does not reveal more information than any other existing FL-HPO approaches. In particular, as far as we know, all existing FL-HPO approaches, e.g., FedEx [Khodak et al. 2021], require parties to share raw HP observations, i.e., HP configurations and their corresponding performance metrics (training loss/validation loss/accuracy). Therefore, FLoRA provides the same level of privacy guarantee as other related FL-HPO approaches.
>
> Secondly, the main goal of this paper is to propose a single-shot FL-HPO approach for optimizing global HP, see (2.5). Although FLoRA does not provide a formal privacy guarantee, it is the first FL-HPO approach that selects HPs **without the need of FL training** (a key difference between single-shot and multi-shot approaches), and hence leads to great **savings of communication cost**.
> We agree with the reviewer that even not sharing the raw data of parties, local HPO results might still reveal private information.
> However, we argue that FLoRA with MPLM and APLM provides extra privacy protection comparing to existing FL-HPO approaches because it only requires parties to share their local loss surfaces not raw HP observations.
> Moreover, FLoRA can be integrated with formal privacy techniques, such as differential privacy, which is one of our future directions.
>
> Finally, the paper [Kusner et al., 2015] cited by the reviewer focused on a different problem in which one party tries to perform HPO locally (minimizing its own loss) and then releases the HP, which is related but not exactly the FL-HPO problem.
>
> References:
>
> * [Li et al 2020] Li, Tian, et al. "Federated optimization in heterogeneous networks." Proceedings of Machine Learning and Systems 2 (2020): 429-450.
>
> * [Karimireddy et al 2020] Karimireddy, Sai Praneeth, et al. "Scaffold: Stochastic controlled averaging for federated learning." International Conference on Machine Learning. PMLR, 2020.
>
> * [Khodak et al. 2021] Khodak, Mikhail, et al. "Federated hyperparameter tuning: Challenges, baselines, and connections to weight-sharing." Advances in Neural Information Processing Systems 34 (2021): 19184-19197.
>
> * [Kusner et al., 2015] Kusner, Matt, et al. "Differentially private Bayesian optimization." International conference on machine learning. PMLR, 2015.

---

> ### Author Response · Authors · 2022-11-11
> **Response Thread (2/5)**
>
> * **Baselines**
> > In the experiments, the proposed method is only compared with a very weak baseline which simply uses the default HP from the package (it is not even optimized by any algorithm)...
> ---------------------------------
>
> We want to point out that FL-HPO is a very new area and the closest work is [Khodak et al. 2021] published in NeurIPS 2021 where they proposed a multi-shot FL-HPO approach applicable to only SGD-based algorithms. In their numerical results, they do not compare to any external baselines but only one natural baseline to run SHA without their approach. In our paper, we have compared FLoRA with $2$ considerably strong external baselines.
>
> In particular, as we have discussed in Section 1.1 (Related Work), there are not a lot of baselines for FL-HPO -- they either focus only on specific ML models (neural networks or linear models) or/and specific HP of those models (such as learning rate or the number of epochs) and **none of them are single-shot**.
> Moreover, *a single FL training is already a very expensive process*. So we focused on the **single-shot** scenario where we are trying to perform FL-HPO but with just a single FL training.
>
> Given these two properties of FLoRA -- (i) agnosticity to ML model type and hyper-parameters, and (ii) single-shot, we were **unable to find any existing baselines** that would be applicable. Hence we considered $2$ strong baselines in our experiments --
>
> (a) A carefully tuned single-shot baseline (we discuss this **Baselines** on Page 7) where we compare the predictive performance achieved. These HPs in the Auto-sklearn package are selected via careful meta-learning [Feurer et al., 2015] and portfolio selection [Fuerer et al., 2020].
>
> (b) A centralized multi-shot baseline involving multiple FL training where we compare the communication savings FLoRA provides over the baseline for the same level of predictive performance.
>
> Furthermore, the reviewer pointed out an interesting "nature baseline" where the algorithm returns the HP that gives the lowest loss among all HP observations from all agents. We want to argue that this "nature baseline" is essentially FLoRA with SGM loss surface where the unified loss surface (line 6 of the Algorithm) is trained on all collected HP/loss pairs. More specifically, let us assume that one uses a random forest as the regressor for SGM. Since we train the regressor only on the collected local HPO results, this regressor can only predict loss with the minimum loss value being the one it has seen in all local HPO runs and selects the corresponding HP configuration as the best HP.
>
> As we can see from our numerical results, in particular Table 4, FLoRA with SGM works fine in cases where parties have IID per-party data distributions. However, it is less robust in heterogeneous cases. This behavior is also supported by our theoretical analysis.
>
> Finally, we want to reiterate that, to the best of our knowledge, existing FL-HPO approaches are either multi-shot approaches that only work for non-tree-based models or they cannot handle algorithmic and model architecture hyperparameters at the same time.
>
> ---------------------------------
> References:
>
> * [Khodak et al. 2021] Khodak, Mikhail, et al. "Federated hyperparameter tuning: Challenges, baselines, and connections to weight-sharing." Advances in Neural Information Processing Systems 34 (2021): 19184-19197.
>
> * [Feurer et al., 2015] Feurer, Matthias, et al. "Efficient and robust automated machine learning." Advances in neural information processing systems 28 (2015).
>
> * [Feurer et al., 2020] Feurer, Matthias, et al. "Auto-sklearn 2.0: The next generation." arXiv preprint arXiv:2007.04074 24 (2020).

---

> ### Author Response · Authors · 2022-11-11
> **Response Thread (3/5)**
>
> * **Real FL systems for numerical experiments**
> > Another important weakness of the empirical evaluation is that the FL algorithm is in fact emulated using centralized training (mentioned at the bottom of page 7). I think this makes the empirical evaluations unrealistic, and may not be able to reflect some unique features/challenges of the FL setting such as heterogeneity.
>
> ---------------------------------
>
> We first want to point out that we use a centralized simulation environment to conduct most of our numerical experiments because we can compute the true regret (w.r.t. best metric obtained via centralized HPO) which is not possible in real FL settings.
> Moreover, we want to emphasize that centralized simulation does not prevent us from demonstrating the effectiveness of FLoRA running on real FL systems for the following reasons: 1) in the simulated FL setting, we can still simulate heterogeneous per-party local data distributions -- see Table 4 and 5 where the data splitting process explicitly introduces heterogeneous per-party data distributions in this simulated FL system; 2) another key difference between a simulated FL environment and a real FL system is communication, but FLoRA is single-shot FL-HPO approach (with minimal communication overhead over a single FL training) and hence its performance will be least affected.
>
> We would also like to highlight Appendix D.9, where we have run our experiments on a real FL system utilizing the IBM FL library [Ludwig et al., 2020] for federated gradient-boosted trees [Ong et al., 2020]. Table 15 shows that FLoRA can improve over baselines in terms of balanced accuracy for several datasets.
>
> ---------------------------------
>
> * **Table 4**
> > Table 4: some of the results are in fact worse than the baseline (which is in fact not so competitive as I discussed above) especially when the number of agents is large, which is worrying for the practical deployment of the proposed method especially in problems with a large number of agents.
> ---------------------------------
>
> We take this opportunity to further clarify the experimental setup and the results in Table 4.
> The goal of this experiment is to stress test the performance of FLoRA under heterogeneous settings. As we have explained in section "Effect of data heterogeneity" on Page 8,  "for each data set, we increase the number of parties $p$ up until each party has at least 100 training samples".
> The increasing number of parties exacerbates two challenges: (i) As the number of parties increases, the training data available on each party decreases, deteriorating the quality of the HP/loss pairs generated on each party, and (ii) The heterogeneity between the per-party data distribution increases with the number of parties, as evidenced by the increasing $\gamma_p$.
>
>
> We want to draw the reviewer's attention to those few cases where FLoRA fails to improve over baselines. First, we note that among the results with worse performances, $\gamma_p$ is relatively large (implying larger heterogeneity in the FL system). This matches our theoretical analysis. Second, FLoRA with SGM fails more often than robust loss surfaces such as MPLM and APLM. As we have explained in the above point (see **Baselines**), SGM is essentially the "natural baseline" the reviewer described and is less robust to data heterogeneity. We have summarized different characteristics of these 4 types of loss surfaces in Table 2.
>
> Finally, as we have highlighted in another set of experiments whose results were presented in Fig. 3a, the performance of FLoRA is also affected by the number of local HPO rounds (i.e., $T$) performed by the parties. We can see from Fig. 3a, the more local HPO runs, the better performance FLoRA has in general. Therefore, considering the results in Table 4 are obtained when $T=100$, FLoRA should exhibit better performance when we increase $T$.
>
> References:
>
> * [Ludwig et al 2020] Ludwig, Heiko, et al. "Ibm federated learning: an enterprise framework white paper v0. 1." arXiv preprint arXiv:2007.10987 (2020).
>
> * [Ong et al. 2020] Ong, Yuya Jeremy, et al. "Adaptive histogram-based gradient boosted trees for federated learning." arXiv preprint arXiv:2012.06670 (2020).

---

> ### Author Response · Authors · 2022-11-11
> **Response Thread (4/5)**
>
> ---------------------------------
>
> * **Design of function $u$**
> > Top paragraph of page 6: how exactly is the uncertainty quantification function $u$ designed? I guess it's supposed to be larger if an HP has been explored by a smaller number of agents
>
> ---------------------------------
>
> The uncertainty function depends on the regressor used to model the loss surface. Gaussian Process Regressors naturally provide uncertainty estimates for predictions. For Random Forest Regressors, uncertainty estimates for predictions have been generated based on the variance of the individual tree predictions in the Random Forest, and we use this in our experiments.
>
> ---------------------------------
> * **Theoretical results**
> > Top paragraph of page 7: it is claimed that the theoretical results imply that "the optimality gap depends only on the HP trials $\theta_t^{(i)}$ that are closest to the optimal HP setting". However, I think this may not be accurate. This is because there is a max over all $\theta \in \Theta$ on the right hand side of equation 2.9, therefore, to make the term $\min_{t \in [T]} d(\theta, \theta_t^{(i)})$ small for all $\theta$, we need to make sure that the HPs $\theta_t^{(i)}$'s are uniformly distributed in the entire domain. Please correct me if my understanding is incorrect here.
> ---------------------------------
>
> The reviewer is correct that the outer $\max$ over $\theta \in \Theta$ and the $\min_{t \in [T]} d(\theta, \theta_t^{(i)})$ term together do imply that a uniformly distributed HPs would provide a tighter bound. This is also validated in our experiment in Figure 3a highlighting that a higher number of local HPO rounds $T$, corresponding to better coverage of the HP space, provides better performance. However, we do not need HPs uniformly distributed over the entire domain. First, it would make the local HPO prohibitively expensive and also increase the communication cost of the aggregation of the HP/loss pairs. Second, given that the HP $\hat{\theta}^\star$ selected by FLoRA is one that minimizes the aggregated loss surface, it is going to be some HP configuration that is close to regions containing "good HPs" for at least some of the parties. In that case, it is better to use the local HPO to focus more on areas of "interest" rather than uniformly spanning the entire HP space. We briefly discuss this point in the last paragraph of Section 2.1 in the paper where we suggest using adaptive HPO schemes to run local HPO.
>
> ---------------------------------
>
>
> * **Multi-shot baselines in Figure 1 and 2**
> > How exactly does the ``Multi-shot baseline'' in Figure 1 and Figure 2 work? This was never explained.
>
> ---------------------------------
>
> We briefly describe "multi-shot" next to Figure 1 (page 2) as *we directly adopt an existing centralized HPO scheme that requires federated training of multiple models and term this a "multi-shot" FL-HPO baseline*. In our experiments, we utilize Hyperopt [Bergstra et al., 2011] as the centralized HPO scheme. We provide a more detailed visualization and discussion of multi-shot FL-HPO in Figure 4b in Appendix A.
>
> References:
>
> * [Bergstra et al., 2011] Bergstra, James, et al. "Algorithms for hyper-parameter optimization." Advances in neural information processing systems 24 (2011).

---

> ### Author Response · Authors · 2022-11-11
> **Response Thread (5/5)**
>
> ---------------------------------
>
> * **Suggestion to improve over local HPO**
> > This may not be a weakness but rather a suggestion. The local HP optimization step for every agent (line 3 of Algorithm 1) may have room for improvement. Since now we have the opportunity to let every agent independently run HP optimization, we can in fact coordinate the local HP optimizations such that they can combine to lead to more accurate global loss surface estimation. For example, you can consider the method of "distributed exploration" from the work of Dai et al. (2021), and let every agent explore only a small local region of the entire search space.
> ---------------------------------
>
>
> We thank the reviewer's great suggestion for improving our proposed algorithm further. We will definitely consider coordination among parties during the HPO process as a future direction. We also want to point out that one of the goals of this paper (FL-HPO Challenges, C3), explained in the first paragraph of Page 2) is to avoid communication overheads. However, coordinating local HP optimizations will incur additional communication costs in FL settings.
>
> The local HPO process in FLoRA is performed in an asynchronous manner, mostly (i) to keep the overall rounds of communication as small as possible, and (ii) to accommodate for the heterogeneous resource availability and data sizes on the different parties, each requiring different amounts of time for the local HPO; coordinating after each local HPO round is more challenging than coordinating once at the end of the local HPO.
>
> This restriction can be removed and we can perform more coordinated local HPOs if the FL setup allows for it.
>
>
> ---------------------------------
>
> * **minor**
> > (minor) Section 1.1, second paragraph: in "Federated Network Architecture Search", I think Network should be Neural.
>
> ---------------------------------
>
> We will update the paper according to the reviewer's suggestions.

---

### Official Review · Reviewer_rh1d · 2022-10-25

**Confidence:** 3
**Correctness:** 4
**Technical Novelty And Significance:** 3
**Empirical Novelty And Significance:** 3
**Recommendation:** 6

**Clarity, Quality, Novelty And Reproducibility:**

Clarity and Quality:

Overall very good.

Novelty:

Overall seems to be very novel.

Reproducibility:

Overall very good and code is provided. Maybe it will be better with an appendix on more details of experiments.

**Strength And Weaknesses:**

Strength:

1: The paper is well motivated. As stated by authors, FL-HPO is very important due to unavailability of global data and limitations on communication and computation budget.

2: The proposed method focuses on general one-shot FL-HPO, which is very challenging and useful. Authors utilize fully independent local HPO and aggregate them into a global loss surface, which is a very novel idea. Further, authors propose several regressor methods to constrict such a loss surface.

3: Authors provide a thorough theoretical analysis on the proposed method.

4: Authors perform extensive experiments on different proposed methods on different models and tasks. In all experiments, the proposed method seems to be better than the baseline.

Weakness:

1: The major concern is that the main baseline, which is simply the default HP configuration in certain package, might not be a strong enough baseline. First of all, such baseline will not adjust HP based on current tasks and datasets (it is more like a zero-shot HPO baseline). I would suggest find other one-shot HPO methods or communication-restricted multi-shot HPO as main baseline, if it is possible.


**Summary Of The Paper:**

Authors address the problem of hyper-parameter optimization (HPO) for federated learning (FL-HPO). Authors propose FLoRA, which (1) enables one-shot FL-HPO: identifying a single set of good hyper-parameters in a single FL training, (2) can be applied to tabular data and any machine learning models. FLoRA is mainly composed of (1) locally hyper-parameter optimization on each client (2) aggregate local hyperparameter into a global loss surface. Authors theoretically characterize the optimality gap of FLoRA for any convex and non-convex loss functions, which explicitly accounts for the heterogeneous nature of the parties’ local data distributions. Authors use  default HP configuration of scikit-learn as the single-shot baseline. Compared with such a baseline, FLoRA shows significant improvement in final accuracy in several datasets and models. Authors further shows their method achieve significant communication savings compared with multi-shot HPO.


**Summary Of The Review:**

I recommend weak accept of this paper. I must say I am not very familiar with FL-HPO and its related literature. I think the overall paper is well written and the proposed method is novel and effective. The only major concern is about the main baseline choice, which might be replaced with other one shot FL-HPO.

---

> ### Author Response · Authors · 2022-11-11
> **Thank you for the review!**
>
> We thank the reviewer for spending time reviewing our work and providing constructive feedback to us. We appreciate that the reviewer recognizes (i) the importance and the challenges of the studied FL-HPO problem and (ii) the thoroughness of our theoretical and empirical analysis of our proposed single-shot algorithm.
> We address the reviewer's comments and concerns as follows:
>
>
> ---------------------------------
>
> * **Baselines**
> > The major concern is that the main baseline, which is simply the default HP configuration in certain package, might not be a strong enough baseline. First of all, such baseline will not adjust HP based on current tasks and datasets (it is more like a zero-shot HPO baseline). I would suggest find other one-shot HPO methods or communication-restricted multi-shot HPO as main baseline, if it is possible.
>
>
> ---------------------------------
>
>
> We want to point out that FL-HPO is a very new area and the closest work is [Khodak et al. 2021] published in NeurIPS 2021 where they proposed a multi-shot FL-HPO approach applicable to only SGD-based algorithms. In their numerical results, they do not compare to any external baselines. In our paper, we have compared FLoRA with $2$ considerably strong external baselines.
>
> In particular, as we have discussed in Section 1.1 (Related Work), there are not a lot of baselines for FL-HPO -- they either focus only on specific ML models (neural networks or linear models) or/and specific HP of those models (such as learning rate or number of epochs), and **none of them are single-shot**.
>
> As we discuss in the last paragraph of Page 1, FL is important to all ML models (ensembles, kernel machines and neural networks) and relevant hyper-parameters (network architecture, ensemble size, kernel bandwidth). To that end, we propose FLoRA which is agnostic to the ML model type and hyperparameters.
>
> Moreover, *a single FL training is already a very expensive process*. So we focus on the **single-shot** scenario where we are trying to perform FL-HPO but with just a single FL training.
>
>
> Given these two properties of FLoRA -- (i) agnosticity to ML model type and hyper-parameters, and (ii) single-shot nature, we were **unable to find any existing baselines** that would be applicable. Hence we consider $2$ strong baselines in our experiments --
>
> (a) A carefully tuned single-shot baseline (we discuss this in **Baselines** on Page 7) where we compare the predictive performance achieved with single-shot FL-HPO. These baseline HPs (obtained from the Auto-sklearn package) are selected via careful meta-learning [Feurer et al., 2015] and portfolio selection [Feurer et al., 2020].
>
> (b) A centralized multi-shot baseline involving multiple FL training where we compare the communication savings FLoRA provides over the baseline for the same level of predictive performance.
>
> References:
> * [Khodak et al. 2021] Khodak, Mikhail, et al. "Federated hyperparameter tuning: Challenges, baselines, and connections to weight-sharing." Advances in Neural Information Processing Systems 34 (2021): 19184-19197.
>
> * [Feurer et al., 2015] Feurer, Matthias, et al. "Efficient and robust automated machine learning." Advances in neural information processing systems 28 (2015).
>
> * [Feurer et al., 2020] Feurer, Matthias, et al. "Auto-sklearn 2.0: The next generation." arXiv preprint arXiv:2007.04074 24 (2020).

---

### Official Review · Reviewer_wq54 · 2022-10-27

**Confidence:** 3
**Correctness:** 4
**Technical Novelty And Significance:** 3
**Empirical Novelty And Significance:** 3
**Recommendation:** 8

**Clarity, Quality, Novelty And Reproducibility:**

(Clarity) This work is well presented.

(Quality) High quality work.

(Novelty) A novel approach for handling HPO in FL, as far as I know.

(Reproducibility) Good.


**Details Of Ethics Concerns:**

N/A.

**Strength And Weaknesses:**

Strength:

1. The problem studied in this work is well-motivated and important in practice.
2. This work provides a neat solution for solving the problem of hyper-parameter optimization (HPO) for federated learning, with only single shot HPO.
3. The theoretical results covers the non-iid setting in FL.
4. Empirically, the proposed approach largely improves upon previous methods in terms of communication overhead.

Weaknesses:
1. (minor) The experiments mainly focus on the non-deep learning approaches and MLP architectures, it might be interesting to include some modern deep network architectures, e.g., ResNet/ViT etc.

Questions:
1. Regarding the loss surface $\hat{\ell}$, under certain assumptions on $f(\theta)$, e.g. $f$ is (strongly) convex, can the upper bound in Eq.(2.9) be further improved?

**Summary Of The Paper:**

This paper studies how to perform hyper-parameter optimization in the context of federated learning, with minimal additional communication overhead. The authors propose a framework ---Federated Loss SuRface Aggregation(FLoRA)--- that requires only single-shot HPO for hyper-parameter optimization for FL. Theoretically, this paper provides an upper bound on the optimality gap, which captures the influence of parties’ data heterogeneity (Non-IID-ness) and the quality of the local HPO approximation. Empirically, the proposed new approach could largely reduce the number of communications while maintains similar model performance.

**Summary Of The Review:**

Given the importance of the problem and new approach for handling such problem, together with strong empirical results and interesting theoretical results, I recommend acceptance.

---

> ### Author Response · Authors · 2022-11-10
> **Thank you for your review!**
>
> We highly appreciate the reviewer for spending time reviewing our work and providing constructive feedback to us. We are also thankful for their strong support for the paper.
> We address the reviewer's comments and concerns as follows:
>
>
> ---------------------------------
>
> * **Modern deep network architecture**
> > (minor) The experiments mainly focus on the non-deep learning approaches and MLP architectures, it might be interesting to include some modern deep network architectures, e.g., ResNet/ViT etc.
>
> ---------------------------------
>
>
>
> We thank the reviewer for this suggestion. Obtaining numerical results on more complicated neural network models, such as CNN, ResNet/Vit, etc., will indeed be interesting. Nonetheless, we want to emphasize the following points regarding our experiments:
>
> 1. As the reviewer pointed out, our numerical experiments have covered both traditional machine learning models and neural networks, i.e., MLP. More importantly, FLoRA is the first FL-HPO solution that can optimize HP configurations for tree-based models, such as gradient-boosted decision trees (HGB), not limited to neural networks and SGD-based algorithms as existing FL-HPO solutions do.
>
> 2. The main goal of our MLP experiments is to demonstrate that FLoRA is the **first** FL-HPO approach that can optimize **both algorithmic and model architecture hyper-parameters** simultaneously.
>
>
> ---------------------------------
>
> * **Theoretical analysis**
> > Regarding the loss surface $\hat \ell$, under certain assumptions on $f(\theta)$, e.g.  $f$ is (strongly) convex, can the upper bound in Eq.(2.9) be further improved?
>
> ---------------------------------
>
>
> The reviewer has raised an interesting point on how to improve our current bound for the optimality gap we considered in (2.8). Due to the page limit of the main paper, we have deferred our discussion on a few possible directions to improve our results in (2.9) to Appendix C, including generalization and relaxation of the Lipschitz continuity assumption, etc.
> We want to point out that the major bottleneck for providing a tighter bound for (2.8) is the property of the loss function and the search space, i.e., the HP space $\Theta$. If we can assume both the loss function and the HP space $\Theta$ are convex, and we can obtain local HPO results on the convex hull or the extreme points of $\Theta$, then we will be able to improve the current bound we have in term of the $\max$ operation and also estimate the required number of local HPO runs (i.e., $T$) for a FL-HPO problem. However, the HP space $\Theta$ is usually not even continuous (many HPs are discrete) not to say convex, we, therefore, did not include such discussion in our main results.
>
> ---------------------------------

---

### Decision · Program_Chairs · 2023-01-20

**Decision:**

Accept: notable-top-25%

**Justification For Why Not Higher Score:**

The used baselines appear weak.

**Justification For Why Not Lower Score:**

Combination of novel problem, interesting approach, good results and theoretical results.

**Metareview: Summary, Strengths And Weaknesses:**

This paper introduces a novel algorithm for hyperparameter optimization (HPO) of federated learning algorithms, which applies to many different algorithms, not only neural networks. It also applies to both algorithmic and architectural hyperparameters. The problem is important and the solution novel, and the paper also presents interesting theoretical results. A common criticism was that the baseline of default parameters in scikit-learn is too naive, but the authors argue that there simply don't exist any possible baselines for the setting they consider.
The authors rebutted effectively about some concerns, and all authors are in favour of accepting the paper. Given the combination of novel problem, interesting approach, good results and theoretical results, I'm recommending the paper to be a spotlight (or even oral if the PCs decide for this with a view of all submissions).

**Note From Pc:**

if the above contains the word "oral" or "spotlight" please see: "oral" presentation means -> notable-top-5% and "spotlight" means -> notable-top-25%. As stated in our emails, we are disassociating presentation type from AC recommendations